# Streaming, Memory Limited Algorithms for Community Detection

**Se-Young. Yun**
MSR-Inria
23 Avenue d'Italie, Paris 75013
seyoung.yun@inria.fr

**Marc Lelarge** [*]
Inria & ENS
23 Avenue d'Italie, Paris 75013
marc.lelarge@ens.fr

**Alexandre Proutiere** [†]
KTH, EE School / ACL
Osquldasv. 10, Stockholm 100-44, Sweden
alepro@kth.se

## Abstract

In this paper, we consider sparse networks consisting of a finite number of non-overlapping communities, i.e. disjoint clusters, so that there is higher density within clusters than across clusters. Both the intra- and inter-cluster edge densities vanish when the size of the graph grows large, making the cluster reconstruction problem nosier and hence difficult to solve. We are interested in scenarios where the network size is very large, so that the adjacency matrix of the graph is hard to manipulate and store. The data stream model in which columns of the adjacency matrix are revealed sequentially constitutes a natural framework in this setting. For this model, we develop two novel clustering algorithms that extract the clusters asymptotically accurately. The first algorithm is *offline*, as it needs to store and keep the assignments of nodes to clusters, and requires a memory that scales linearly with the network size. The second algorithm is *online*, as it may classify a node when the corresponding column is revealed and then discard this information. This algorithm requires a memory growing sub-linearly with the network size. To construct these efficient streaming memory-limited clustering algorithms, we first address the problem of clustering with partial information, where only a small proportion of the columns of the adjacency matrix is observed and develop, for this setting, a new spectral algorithm which is of independent interest.

## 1 Introduction

Extracting clusters or communities in networks have numerous applications and constitutes a fundamental task in many disciplines, including social science, biology, and physics. Most methods for clustering networks assume that pairwise "interactions" between nodes can be observed, and that from these observations, one can construct a graph which is then partitioned into clusters. The resulting graph partitioning problem can be typically solved using spectral methods [1, 3, 5, 6, 12], compressed sensing and matrix completion ideas [2, 4], or other techniques [10].

A popular model and benchmark to assess the performance of clustering algorithms is the Stochastic Block Model (SBM) [9], also referred to as the planted partition model. In the SBM, it is assumed

---

[*]Work performed as part of MSR-INRIA joint research centre. M.L. acknowledges the support of the French Agence Nationale de la Recherche (ANR) under reference ANR-11-JS02-005-01 (GAP project).

[†]A. Proutiere's research is supported by the ERC FSA grant, and the SSF ICT-Psi project.

that the graph to partition has been generated randomly, by placing an edge between two nodes with probability $p$ if the nodes belong to the same cluster, and with probability $q$ otherwise, with $q < p$. The parameters $p$ and $q$ typically depends on the network size $n$, and they are often assumed to tend to 0 as $n$ grows large, making the graph sparse. This model has attracted a lot of attention recently. We know for example that there is a phase transition threshold for the value of $\frac{(p-q)^2}{p+q}$. If we are below the threshold, no algorithm can perform better than the algorithm randomly assigning nodes to clusters [7, 14], and if we are above the threshold, it becomes indeed possible to beat the naive random assignment algorithm [11]. A necessary and sufficient condition on $p$ and $q$ for the existence of clustering algorithms that are asymptotically accurate (meaning that the proportion of misclassified nodes tends to 0 as $n$ grows large) has also been identified [15]. We finally know that spectral algorithms can reconstruct the clusters asymptotically accurately as soon as this is at all possible, i.e., they are in a sense optimal.

We focus here on scenarios where the network size can be extremely large (online social and biological networks can, already today, easily exceed several hundreds of millions of nodes), so that the adjacency matrix $A$ of the corresponding graph can become difficult to manipulate and store. We revisit network clustering problems under memory constraints. Memory limited algorithms are relevant in the streaming data model, where observations (i.e. parts of the adjacency matrix) are collected sequentially. We assume here that the columns of the adjacency matrix $A$ are revealed one by one to the algorithm. An arriving column may be stored, but the algorithm cannot request it later on if it was not stored. The objective of this paper is to determine how the memory constraints and the data streaming model affect the fundamental performance limits of clustering algorithms, and how the latter should be modified to accommodate these restrictions. Again to address these questions, we use the stochastic block model as a performance benchmark. Surprisingly, we establish that when there exists an algorithm with unlimited memory that asymptotically reconstruct the clusters accurately, then we can devise an asymptotically accurate algorithm that requires a memory scaling linearly in the network size $n$, except if the graph is extremely sparse. This claim is proved for the SBM with parameters $p = a\frac{f(n)}{n}$ and $q = b\frac{f(n)}{n}$, with constants $a > b$, under the assumption that $\log n \ll f(n)$. For this model, unconstrained algorithms can accurately recover the clusters as soon as $f(n) = \omega(1)$ [15], so that the gap between memory-limited and unconstrained algorithms is rather narrow. We further prove that the proposed algorithm reconstruct the clusters accurately before collecting all the columns of the matrix $A$, i.e., it uses *less* than one pass on the data. We also propose an online streaming algorithm with sublinear memory requirement. This algorithm output the partition of the graph in an online fashion after a group of columns arrives. Specifically, if $f(n) = n^\alpha$ with $0 < \alpha < 1$, our algorithm requires as little as $n^\beta$ memory with $\beta > \max\left(1 - \alpha, \frac{2}{3}\right)$. To the best of our knowledge, our algorithm is the first sublinear streaming algorithm for community detection. Although streaming algorithms for clustering data streams have been analyzed [8], the focus in this theoretical computer science literature is on worst case graphs and on approximation performance which is quite different from ours.

To construct efficient streaming memory-limited clustering algorithms, we first address the problem of clustering with *partial information*. More precisely, we assume that a proportion $\gamma$ (that may depend on $n$) of the columns of $A$ is available, and we wish to classify the nodes corresponding to these columns, i.e., the observed nodes. We show that a necessary and sufficient condition for the existence of asymptotically accurate algorithms is $\sqrt{\gamma}f(n) = \omega(1)$. We also show that to classify the observed nodes efficiently, a clustering algorithm must exploit the information provided by the edges between observed and unobserved nodes. We propose such an algorithm, which in turn, constitutes a critical building block in the design of memory-limited clustering schemes.

To our knowledge, this paper is the first to address the problem of community detection in the streaming model, and with memory constraints. Note that PCA has been recently investigated in the streaming model and with limited memory [13]. Our model is different, and to obtain efficient clustering algorithms, we need to exploit its structure.

## 2   Models and Problem Formulation

We consider a network consisting of a set $V$ of $n$ nodes. $V$ admits a hidden partition of $K$ non-overlapping subsets $V_1, \ldots, V_K$, i.e., $V = \bigcup_{k=1}^{K} V_k$. The size of community or cluster $V_k$ is $\alpha_k n$ for some $\alpha_k > 0$. Without loss of generality, let $\alpha_1 \leq \alpha_2 \leq \cdots \leq \alpha_K$. We assume that when the

network size $n$ grows large, the number of communities $K$ and their relative sizes are kept fixed. To recover the hidden partition, we have access to a $n \times n$ symmetric random binary matrix $A$ whose entries are independent and satisfy: for all $v, w \in V$, $\mathbb{P}[A_{vw} = 1] = p$ if $v$ and $w$ are in the same cluster, and $\mathbb{P}[A_{vw} = 1] = q$ otherwise, with $q < p$. This corresponds to the celebrated Stochastic Block Model (SBM). If $A_{vw} = 1$, we say that nodes $v$ and $w$ are connected, or that there is an edge between $v$ and $w$. $p$ and $q$ typically depend on the network size $n$. To simplify the presentation, we assume that there exists a function $f(n)$, and two constants $a > b$ such that $p = a\frac{f(n)}{n}$ and $q = b\frac{f(n)}{n}$. This assumption on the specific scaling of $p$ and $q$ is not crucial, and most of the results derived in this paper hold for more general $p$ and $q$ (as it can be seen in the proofs). For an algorithm $\pi$, we denote by $\varepsilon^{\pi}(n)$ the proportion of nodes that are misclassified by this algorithm. We say that $\pi$ is asymptotically accurate if $\lim_{n \to \infty} \mathbb{E}[\varepsilon^{\pi}(n)] = 0$. Note that in our setting, if $f(n) = O(1)$, there is a non-vanishing fraction of isolated nodes for which no algorithm will perform better than a random guess. In particular, no algorithm can be asymptotically accurate. Hence, we assume that $f(n) = \omega(1)$, which constitutes a necessary condition for the graph to be asymptotically connected, i.e., the largest connected component to have size $n - o(n)$.

In this paper, we address the problem of reconstructing the clusters from specific observed entries of $A$, and under some constraints related to the memory available to process the data and on the way observations are revealed and stored. More precisely, we consider the two following problems.

**Problem 1. Clustering with partial information.** We first investigate the problem of detecting communities under the assumption that the matrix $A$ is partially observable. More precisely, we assume that a proportion $\gamma$ (that typically depend on the network size $n$) of the columns of $A$ are known. The $\gamma n$ observed columns are selected uniformly at random among all columns of $A$. Given these observations, we wish to determine the set of parameters $\gamma$ and $f(n)$ such that there exists an asymptotically accurate clustering algorithm.

**Problem 2. Clustering in the streaming model and under memory constraints.** We are interested here in scenarios where the matrix $A$ cannot be stored entirely, and restrict our attention to algorithms that require memory less than $M$ bits. Ideally, we would like to devise an asymptotically accurate clustering algorithm that requires a memory $M$ scaling linearly or sub-linearly with the network size $n$. In the streaming model, we assume that at each time $t = 1, \ldots, n$, we observe a column $A_v$ of $A$ uniformly distributed over the set of columns that have not been observed before $t$. The column $A_v$ may be stored at time $t$, but we cannot request it later on if it has not been explicitly stored. The problem is to design a clustering algorithm $\pi$ such that in the streaming model, $\pi$ is asymptotically accurate, and requires less than $M$ bits of memory. We distinguish offline clustering algorithms that must store the mapping between all nodes and their clusters (here $M$ has to scale linearly with $n$), and online algorithms that may classify the nodes when the corresponding columns are observed, and then discard this information (here $M$ could scale sub-linearly with $n$).

## 3 Clustering with Partial Information

In this section, we solve Problem 1. In what follows, we assume that $\gamma n = \omega(1)$, which simply means that the number of observed columns of $A$ grows large when $n$ tends to $\infty$. However we are typically interested in scenarios where the proportion of observed columns $\gamma$ tends to 0 as the network size grows large. Let $(A_v, v \in V^{(g)})$ denote the observed columns of $A$. $V^{(g)}$ is referred to as the set of *green* nodes and we denote by $n^{(g)} = \gamma n$ the number of green nodes. $V^{(r)} = V \setminus V^{(g)}$ is referred to as the set of *red* nodes. Note that we have no information about the connections among the red nodes. For any $k = 1, \ldots, K$, let $V_k^{(g)} = V^{(g)} \cap V_k$, and $V_k^{(r)} = V^{(r)} \cap V_k$. We say that a clustering algorithm $\pi$ classifies the green nodes asymptotically accurately, if the proportion of misclassified green nodes, denoted by $\varepsilon^{\pi}(n^{(g)})$, tends to 0 as the network size $n$ grows large.

### 3.1 Necessary Conditions for Accurate Detection

We first derive necessary conditions for the existence of asymptotically accurate clustering algorithms. As it is usual in this setting, the hardest model to estimate (from a statistical point of view) corresponds to the case of two clusters of equal sizes (see Remark 3 below). Hence, we state our information theoretic lower bounds, Theorems 1 and 2, for the special case where $K = 2$, and

$\alpha_1 = \alpha_2$. Theorem 1 states that if the proportion of observed columns $\gamma$ is such that $\sqrt{\gamma}f(n)$ tends to 0 as $n$ grows large, then no clustering algorithm can perform better than the naive algorithm that assigns nodes to clusters randomly.

**Theorem 1** *Assume that $\sqrt{\gamma}f(n) = o(1)$. Then under any clustering algorithm $\pi$, the expected proportion of misclassified green nodes tends to 1/2 as $n$ grows large, i.e., $\lim_{n\to\infty} \mathbb{E}[\varepsilon^\pi(n^{(g)})] = 1/2$.*

Theorem 2 (i) shows that this condition is tight in the sense that as soon as there exists a clustering algorithm that classifies the green nodes asymptotically accurately, then we need to have $\sqrt{\gamma}f(n) = \omega(1)$. Although we do not observe the connections among red nodes, we might ask to classify these nodes through their connection patterns with green nodes. Theorem 2 (ii) shows that this is possible only if $\gamma f(n)$ tends to infinity as $n$ grows large.

**Theorem 2** *(i) If there exists a clustering algorithm that classifies the green nodes asymptotically accurately, then we have: $\sqrt{\gamma}f(n) = \omega(1)$.*
*(ii) If there exists an asymptotically accurate clustering algorithm (i.e., classifying all nodes asymptotically accurately), then we have: $\gamma f(n) = \omega(1)$.*

**Remark 3** *Theorems 1 and 2 might appear restrictive as they only deal with the case of two clusters of equal sizes. This is not the case as we will provide in the next section an algorithm achieving the bounds of Theorem 2 (i) and (ii) for the general case (with a finite number $K$ of clusters of possibly different sizes). In other words, Theorems 1 and 2 translates directly in minimax lower bounds thanks to the results we obtain in Section 3.2.*

Note that as soon as $\gamma f(n) = \omega(1)$ (i.e. the mean degree in the observed graph tends to infinity), then standard spectral method applied on the squared matrix $A^{(g)} = (A_{vw}, v, w \in V^{(g)})$ will allow us to classify asymptotically accurately the green nodes, i.e., taking into account only the graph induced by the green vertices is sufficient. However if $\gamma f(n) = o(1)$ then no algorithm based on the induced graph only will be able to classify the green nodes. Theorem 2 shows that in the range of parameters $1/f(n)^2 \ll \gamma \ll 1/f(n)$, it is impossible to cluster asymptotically accurately the red nodes but the question of clustering the green nodes is left open.

## 3.2 Algorithms

In this section, we deal with the general case and assume that the number $K$ of clusters (of possibly different sizes) is known. There are two questions of interest: clustering green and red nodes. It seems intuitive that red nodes can be classified only if we are able to first classify green nodes. Indeed as we will see below, once the green nodes have been classified, an easy greedy rule is optimal for the red nodes.

**Classifying green nodes.** Our algorithm to classify green nodes rely on spectral methods. Note that as suggested above, in the regime $1/f(n)^2 \ll \gamma \ll 1/f(n)$, any efficient algorithm needs to exploit the observed connections between green and red nodes. We construct such an algorithm below. We should stress that our algorithm does not require to know or estimate $\gamma$ or $f(n)$.

When from the observations, a red node $w \in V^{(r)}$ is connected to at most a single green node, i.e., if $\sum_{v \in V^{(g)}} A_{vw} \leq 1$, this red node is useless in the classification of green nodes. On the contrary, when a red node is connected to two green nodes, say $v_1$ and $v_2$ ($A_{v_1 w} = 1 = A_{v_2 w}$), we may infer that the green nodes $v_1$ and $v_2$ are likely to be in the same cluster. In this case, we say that there is an *indirect edge* between $v_1$ and $v_2$.

To classify the green nodes, we will use the matrix $A^{(g)} = (A_{vw})_{v,w \in V^{(g)}}$, as well as the graph of indirect edges. However this graph is statistically different from the graphs arising in the classical stochastic block model. Indeed, when a red node is connected to three or more green nodes, then the presence of indirect edges between these green nodes are not statistically independent. To circumvent this difficulty, we only consider indirect edges created through red nodes connected to exactly two green nodes. Let $V^{(i)} = \{v : v \in V^{(r)} \text{ and } \sum_{w \in V^{(g)}} A_{wv} = 2\}$. We denote by $A'$ the $(n^{(g)} \times n^{(g)})$ matrix reporting the number of such indirect edges between pairs of green nodes: for all $v, w \in V^{(g)}$, $A'_{vw} = \sum_{z \in V^{(i)}} A_{vz} A_{wz}$.

---

**Algorithm 1** Spectral method with indirect edges

---

**Input:** $A \in \{0,1\}^{|V| \times |V^{(g)}|}$, $V$, $V^{(g)}$, $K$

$V^{(r)} \leftarrow V \setminus V^{(g)}$

$V^{(i)} \leftarrow \{v : v \in V^{(r)} \text{ and } \sum_{w \in V^{(g)}} A_{wv} = 2\}$

$A^{(g)} \leftarrow (A_{vw})_{v,w \in V^{(g)}}$ and $A' \leftarrow (A'_{vw} = \sum_{z \in V^{(i)}} A_{vz} A_{wz})_{v,w \in V^{(g)}}$

$\hat{p}^{(g)} \leftarrow \frac{\sum_{v,w \in V^{(g)}} A^{(g)}_{vw}}{|V^{(g)}|^2}$ and $\hat{p}' \leftarrow \frac{\sum_{v,w \in V^{(g)}} A'_{vw}}{|V^{(g)}|^2}$

$Q^{(g)}, \sigma_K^{(g)}, \Gamma^{(g)} \leftarrow \text{Approx}(A^{(g)}, \hat{p}^{(g)}, V^{(g)}, K)$ and $Q', \sigma'_K, \Gamma' \leftarrow \text{Approx}(A', \hat{p}', V^{(g)}, K)$

**if** $\frac{\sigma_K^{(g)}}{\sqrt{|V^{(g)}|\hat{p}^{(g)}}} \cdot 1_{\{|V^{(g)}|\hat{p}^{(g)} \geq 50\}} \geq \frac{\sigma'_K}{\sqrt{|V^{(g)}|\hat{p}'}} \cdot 1_{\{|V^{(g)}|\hat{p}' \geq 50\}}$ **then**

    $(S_k)_{1 \leq k \leq K} \leftarrow \text{Detection}(Q^{(g)}, \Gamma^{(g)}, K)$

    Randomly place nodes in $V^{(g)} \setminus \Gamma^{(g)}$ to partitions $(S_k)_{k=1,\dots,K}$

**else**

    $(S_k)_{1 \leq k \leq K} \leftarrow \text{Detection}(Q', \Gamma', K)$

    Randomly place nodes in $V^{(g)} \setminus \Gamma'$ to partitions $(S_k)_{k=1,\dots,K}$

**end if**

**Output:** $(S_k)_{1 \leq k \leq K}$,

---

Our algorithm to classify the green nodes consists in the following steps:

Step 1. Construct the indirect edge matrix $A'$ using red nodes connected to two green nodes only.

Step 2. Perform a spectral analysis of matrices $A^{(g)}$ and $A'$ as follows: first trim $A^{(g)}$ and $A'$ (to remove nodes with too many connections), then extract their $K$ largest eigenvalues and the corresponding eigenvectors.

Step 3. Select the matrix $A^{(g)}$ or $A'$ with the largest normalized $K$-th largest eigenvalue.

Step 4. Construct the $K$ clusters $V_1^{(g)}, \dots, V_K^{(g)}$ based on the eigenvectors of the matrix selected in the previous step.

The detailed pseudo-code of the algorithm is presented in Algorithm 1. Steps 2 and 4 of the algorithm are standard techniques used in clustering for the SBM, see e.g. [5]. The algorithms involved in these Steps are presented in the supplementary material (see Algorithms 4, 5, 6). Note that to extract the $K$ largest eigenvalues and the corresponding eigenvectors of a matrix, we use the power method, which is memory-efficient (this becomes important when addressing Problem 2). Further observe that in Step 3, the algorithm exploits the information provided by the red nodes: it selects, between the direct edge matrix $A^{(g)}$ and the indirect edge matrix $A'$, the matrix whose spectral properties provide more accurate information about the $K$ clusters. This crucial step is enough for the algorithm to classify the green nodes asymptotically accurately whenever this is at all possible, as stated in the following theorem:

**Theorem 4** *When $\sqrt{\gamma} f(n) = \omega(1)$, Algorithm 1 classifies the green nodes asymptotically accurately.*

In view of Theorem 2 (i), our algorithm is optimal. It might be surprising to choose one of the matrix $A^{(g)}$ or $A'$ and throw the information contained in the other one. But the following simple calculation gives the main idea. To simplify, consider the case $\gamma f(n) = o(1)$ so that we know that the matrix $A^{(g)}$ alone is not sufficient to find the clusters. In this case, it is easy to see that the matrix $A'$ alone allows to classify as soon as $\sqrt{\gamma} f(n) = \omega(1)$. Indeed, the probability of getting an indirect edge between two green nodes is of the order $(a^2 + b^2) f(n)^2/(2n)$ if the two nodes are in the same clusters and $abf(n)^2/n$ if they are in different clusters. Moreover the graph of indirect edges has the same statistics as a SBM with these probabilities of connection. Hence standard results show that spectral methods will work as soon as $\gamma f(n)^2$ tends to infinity, i.e. the mean degree in the observed graph of indirect edges tends to infinity. In the case where $\gamma f(n)$ is too large (indeed $\gg \ln(f(n))$), then the graph of indirect edges becomes too sparse for $A'$ to be useful. But in this regime, $A^{(g)}$ allows to classify the green nodes. This argument gives some intuitions about the full proof of Theorem 4 which can be found in the Appendix.

---
**Algorithm 2** Greedy selections
---
**Input:** $A \in \{0,1\}^{|V| \times |V^{(g)}|}$, $V$, $V^{(g)}$, $(S_k^{(g)})_{1 \leq k \leq K}$.
$V^{(r)} \leftarrow V \setminus V^{(g)}$ and $S_k \leftarrow S_k^{(g)}$, for all $k$
 **for** $v \in V^{(r)}$ **do**
  Find $k^\star = \arg\max_k \{\sum_{w \in S_k^{(g)}} A_{vw}/|S_k^{(g)}|\}$ (tie broken uniformly at random)
  $S_{k^\star} \leftarrow S_{k^\star} \cup \{v\}$
 **end for**
 **Output:** $(S_k)_{1 \leq k \leq K}$.
---

An attractive feature of our Algorithm 1 is that it does not require any parameter of the model as input except the number of clusters $K$. In particular, our algorithm selects automatically the best matrix among $A'$ and $A^{(g)}$ based on their spectral properties.

**Classifying red nodes.** From Theorem 2 (ii), in order to classify red nodes, we need to assume that $\gamma f(n) = \omega(1)$. Under this assumption, the green nodes are well classified under Algorithm 1. To classify the red nodes accurately, we show that it is enough to greedily assign these nodes to the clusters of green nodes identified using Algorithm 1. More precisely, a red node $v$ is assigned to the cluster that maximizes the number of observed edges between $v$ and the green nodes of this cluster. The pseudo-code of this procedure is presented in Algorithm 2.

**Theorem 5** *When $\gamma f(n) = \omega(1)$, combining Algorithms 1 and 2 yields an asymptotically accurate clustering algorithm.*

Again in view of Theorem 2 (ii), our algorithm is optimal. To summarize our results about Problem 1, i.e., clustering with partial information, we have shown that:
(a) If $\gamma \ll 1/f(n)^2$, no clustering algorithm can perform better than the naive algorithm that assigns nodes to clusters randomly (in the case of two clusters of equal sizes).
(b) If $1/f(n)^2 \ll \gamma \ll 1/f(n)$, Algorithm 1 classifies the green nodes asymptotically accurately, but no algorithm can classify the red nodes asymptotically accurately.
(c) If $1/f(n) \ll \gamma$, the combination of Algorithm 1 and Algorithm 2 classifies all nodes asymptotically accurately.

## 4 Clustering in the Streaming Model under Memory Constraints

In this section, we address Problem 2 where the clustering problem has additional constraints. Namely, the memory available to the algorithm is limited (memory constraints) and each column $A_v$ of $A$ is observed only once, hence if it is not stored, this information is lost (streaming model).

In view of previous results, when the entire matrix $A$ is available (i.e. $\gamma = 1$) and when there is no memory constraint, we know that a necessary and sufficient condition for the existence of asymptotically accurate clustering algorithms is that $f(n) = \omega(1)$. Here we first devise a clustering algorithm adapted to the streaming model and using a memory scaling linearly with $n$ that is asymptotically accurate as soon as $\log(n) \ll f(n)$. Algorithms 1 and 2 are the building blocks of this algorithm, and its performance analysis leverages the results of previous section. We also show that our algorithm does not need to sequentially observe all columns of $A$ in order to accurately reconstruct the clusters. In other words, the algorithm uses strictly less than one pass on the data and is asymptotically accurate.

Clearly if the algorithm is asked (as above) to output the full partition of the network, it will require a memory scaling linearly with $n$, the size of the output. However, in the streaming model, we can remove this requirement and the algorithm can output the full partition sequentially similarly to an online algorithm (however our algorithm is not required to take an irrevocable action after the arrival of each column but will classify nodes after a group of columns arrives). In this case, the memory requirement can be sublinear. We present an algorithm with a memory requirement which depends on the density of the graph. In the particular case where $f(n) = n^\alpha$ with $0 < \alpha < 1$, our algorithm requires as little as $n^\beta$ bits of memory with $\beta > \max\left(1 - \alpha, \frac{2}{3}\right)$ to accurately cluster the nodes. Note that when the graph is very sparse ($\alpha \approx 0$), then the community detection is a hard statistical task and the algorithm needs to gather a lot of columns so that the memory requirement is quite

---

**Algorithm 3** Streaming offline

---

**Input:** $\{A_1, \ldots, A_T\}$, $p$, $V$, $K$

**Initial:** $N \leftarrow n \times K$ matrix filled with zeros and $B \leftarrow \frac{nh(n)}{\min\{np, n^{1/3}\}\log n}$

**Subsampling:** $A_t \leftarrow$ Randomly erase entries of $A_t$ with probability $\max\{0, 1 - \frac{n^{1/3}}{np}\}$

**for** $\tau = 1$ **to** $\lfloor \frac{T}{B} \rfloor$ **do**
    $A^{(B)} \leftarrow n \times B$ matrix where $i$-th column is $A_{i+(\tau-1)B}$
    $(S_k^{(\tau)}) \leftarrow$ Algorithm 1 $(A^{(B)}, V, \{(\tau-1)B + 1, \ldots, \tau B\}, K)$
    **if** $\tau = 1$ **then**
        $\hat{V}_k \leftarrow S_k^{(1)}$ for all $k$ and $N_{v,k} \leftarrow \sum_{w \in S_k^{(1)}} A_{wv}$ for all $v \in V$ and $k$
    **else**
        $\hat{V}_{s(k)} \leftarrow \hat{V}_{s(k)} \cup S_k^{(\tau)}$ for all $k$ where $s(k) = \arg\max_{1 \le i \le K} \frac{\sum_{v \in \hat{V}_i} \sum_{w \in S_k^{(\tau)}} A_{vw}}{|\hat{V}_i||S_k^{(\tau)}|}$
        $N_{v,s(k)} \leftarrow N_{v,s(k)} + \sum_{w \in S_k^{(\tau)}} A_{wv}$ for all $v \in V$ and $k$
    **end if**
**end for**
**Greedy improvement :** $\bar{V}_k \leftarrow \{v : k = \arg\max_{1 \le i \le K} \frac{N_{v,i}}{|\hat{V}_i|}\}$ for all $k$
**Output:** $(\bar{V}_k)_{1 \le k \le K}$,

---

high ($\beta \approx 1$). As $\alpha$ increases, the graph becomes denser and the statistical task easier. As a result, our algorithm needs to look at smaller blocks of columns and the memory requirement decreases. However, for $\alpha \ge 1/3$, although the statistical task is much easier, our algorithm hits its memory constraint and in order to store blocks with sufficiently many columns, it needs to subsample each column. As a result, the memory requirement of our algorithm does not decrease for $\alpha \ge 1/3$.

The main idea of our algorithms is to successively treat *blocks* of $B$ consecutive arriving columns. Each column of a block is stored in the memory. After the last column of a block arrives, we apply Algorithm 1 to classify the corresponding nodes accurately, and we then merge the obtained clusters with the previously identified clusters. In the online version, the algorithm can output the partition of the block and in the offline version, it stores this result. We finally remove the stored columns, and proceed with the next block. For the offline algorithm, after a total of $T$ observed columns, we apply Algorithm 2 to classify the remaining nodes so that $T$ can be less than $n$. The pseudo-code of the offline algorithm is presented in Algorithm 3. Next we discuss how to tune $B$ and $T$ so that the classification is asymptotically accurate, and we compute the required memory to implement the algorithm.

**Block size.** We denote by $B$ the size of a block. Let $h(n)$ be such that the block size is $B = \frac{h(n)n}{f(n)\log(n)}$. Let $\bar{f}(n) = \min\{f(n), n^{1/3}\}$ which represents the order of the number of positive entries of each column after the subsampling process. According to Theorem 4 (with $\gamma = B/n$), to accurately classify the nodes arrived in a block, we just need that $\frac{B}{n}\bar{f}(n)^2 = \omega(1)$, which is equivalent to $h(n) = \omega(\frac{\log(n)}{\min\{f(n), n^{1/3}\}})$. Now the merging procedure that combines the clusters found analyzing the current block with the previously identified clusters uses the number of connections between the nodes corresponding to the columns of the current block to the previous clusters. The number of these connections must grow large as $n$ tends to $\infty$ to ensure the accuracy of the merging procedure. Since the number of these connections scales as $B^2 \frac{\bar{f}(n)}{n}$, we need that $h(n)^2 = \omega(\min\{f(n), n^{1/3}\}\frac{\log(n)^2}{n})$. Note that this condition is satisfied as long as $h(n) = \omega(\frac{\log(n)}{\min\{f(n), n^{1/3}\}})$.

**Total number of columns for the offline algorithm.** To accurately classify the nodes whose columns are not observed, we will show that we need the total number of observed columns $T$ to satisfy $T = \omega(\frac{n}{\min\{f(n), n^{1/3}\}})$ (which is in agreement with Theorem 5).

**Required memory for the offline algorithm.** To store the columns of a block, we need $\Theta(nh(n))$ bits. To store the previously identified clusters, we need at most $\log_2(K)n$ bits, and we can store the number of connections between the nodes corresponding to the columns of the current block to the previous clusters using a memory linearly scaling with $n$. Finally, to execute Algorithm 1, the

---

**Algorithm 4** Streaming online

---

**Input:** $\{A_1, \ldots, A_n\}, p, V, K$

**Initial:** $B \leftarrow \frac{nh(n)}{\min\{np, n^{1/3}\} \log n}$ and $\tau^\star = \lfloor \frac{T}{B} \rfloor$

**Subsampling:** $A_t \leftarrow$ Randomly erase entries of $A_t$ with probability $\max\{0, 1 - \frac{n^{1/3}}{np}\}$

**for** $\tau = 1$ **to** $\tau^\star$ **do**

    $A^{(B)} \leftarrow n \times B$ matrix where $i$-th column is $A_{i+(\tau-1)B}$

    $(S_k)_{1 \leq k \leq K} \leftarrow$ Algorithm 1 $(A^{(B)}, V, \{(\tau-1)B+1, \ldots, \tau B\}, K)$

    **if** $\tau = 1$ **then**

        $\hat{V}_k \leftarrow S_k$ for all $k$

        **Output at** $B$**:** $(S_k)_{1 \leq k \leq K}$

    **else**

        $s(k) \leftarrow \arg\max_{1 \leq i \leq K} \frac{\sum_{v \in \hat{V}_i} \sum_{w \in S_k} A_{vw}}{|\hat{V}_i||S_k|}$ for all $k$

        **Output at** $\tau B$**:** $(S_{s(k)})_{1 \leq k \leq K}$

    **end if**

**end for**

---

power method used to perform the SVD (see Algorithm 5) requires the same amount of bits than that used to store a block of size $B$. In summary, the required memory is $M = \Theta(nh(n) + n)$.

**Theorem 6** *Assume that* $h(n) = \omega(\frac{\log(n)}{\min\{f(n), n^{1/3}\}})$ *and* $T = \omega(\frac{n}{\min\{f(n), n^{1/3}\}})$. *Then with* $M = \Theta(nh(n) + n)$ *bits, Algorithm 3, with block size* $B = \frac{h(n)n}{\min\{f(n), n^{1/3}\} \log(n)}$ *and acquiring the $T$ first columns of $A$, outputs clusters* $\hat{V}_1, \ldots, \hat{V}_K$ *such that with high probability, there exists a permutation $\sigma$ of $\{1, \ldots, K\}$ such that:* $\frac{1}{n} \left| \bigcup_{1 \leq k \leq K} \hat{V}_k \setminus V_{\sigma(k)} \right| = O\left(\exp(-cT \frac{\min\{f(n), n^{1/3}\}}{n})\right)$ *with a constant $c > 0$.*

Under the conditions of the above theorem, Algorithm 3 is asymptotically accurate. Now if $f(n) = \omega(\log(n))$, we can choose $h(n) = 1$. Then Algorithm 3 classifies nodes accurately and uses a memory linearly scaling with $n$. Note that increasing the number of observed columns $T$ just reduces the proportion of misclassified nodes. For example, if $f(n) = \log(n)^2$, with high probability, the proportion of misclassified nodes decays faster than $1/n$ if we acquire only $T = n/\log(n)$ columns, whereas it decays faster than $\exp(-\log(n)^2)$ if all columns are observed.

Our online algorithm is a slight variation of the offline algorithm. Indeed, it deals with the first block exactly in the same manner and keeps in memory the partition of this first block. It then handles the successive blocks as the first block and merges the partition of these blocks with those of the first block as done in the offline algorithm for the second block. Once this is done, the online algorithm just throw all the information away except the partition of the first block.

**Theorem 7** *Assume that* $h(n) = \omega(\frac{\log(n)}{\min\{f(n), n^{1/3}\}})$, *then Algorithm 4 with block size* $B = \frac{h(n)n}{\min\{f(n), n^{1/3}\} \log n}$ *is asymptotically accurate (i.e., after one pass, the fraction of misclassified nodes vanishes) and requires* $\Theta(nh(n))$ *bits of memory.*

## 5 Conclusion

We introduced the problem of community detection with partial information, where only an induced subgraph corresponding to a fraction of the nodes is observed. In this setting, we gave a necessary condition for accurate reconstruction and developed a new spectral algorithm which extracts the clusters whenever this is at all possible. Building on this result, we considered the streaming, memory limited problem of community detection and developed algorithms able to asymptotically reconstruct the clusters with a memory requirement which is linear in the size of the network for the offline version of the algorithm and which is sublinear for its online version. To the best of our knowledge, these algorithms are the first community detection algorithms in the data stream model. The memory requirement of these algorithms is non-increasing in the density of the graph and determining the optimal memory requirement is an interesting open problem.

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
