[Supplementary Material]

# Streaming, Memory Limited Algorithms for Community Detection: Supplementary Material

## A    Algorithms

We present below three algorithms that constitute building blocks of the main algorithms presented in the paper.

---

**Algorithm 4** Approx $(A, \hat{p}, V, K)$

---

**Input:** $A, \hat{p}, V, K$
$\ell^\star \leftarrow \max\{1, \lfloor |V| \exp(-|V|\hat{p}) \rfloor\})$
**for** $v \in V$ **do**
    $x_v \leftarrow \sum_{w \in V} A_{vw}$
**end for**
$x^\star \leftarrow \ell^\star$-th largest $x_v$
$\Gamma \leftarrow \{v | x_v \leq x^\star, \ v \in V\}$
$\bar{A} \leftarrow (A_{vw})_{v,w \in \Gamma}$
$(Q, \sigma_K) \leftarrow$ Power Method $(\bar{A}, \Gamma, K)$ (Algorithm 5)
**Output:** $(Q, \sigma_K, \Gamma)$

---

**Algorithm 5** Power Method $(A, V, K)$

---

**Input:** $A, V, K$
**Initial:** $Q_0 \leftarrow$ Randomly choose $K$ orthonormal vectors and $\tau^\star = \lceil \log |V| \rceil$
**for** $\tau = 1$ **to** $\tau^\star$ **do**
    $AQ_{\tau-1} = Q_\tau R_\tau$
**end for**
$\sigma_K \leftarrow K$-th largest singular value of $R_{\tau^\star}$
**Output:** $(Q_{\tau^\star}, \sigma_K)$

---

**Algorithm 6** Detection $(Q, V, K)$

---

**Input:** $Q, V, K$ (let $Q_v$ denote the low of $Q$ corresponding to $v$)
**for** $i = 1$ **to** $\log |V|$ **do**
   $X_{i,v} \leftarrow \{w \in V : \|Q_w - Q_v\|^2 \leq \frac{i}{|V| \log |V|}\}$
   $T_{i,0} \leftarrow \emptyset$
   **for** $k = 1$ **to** $K$ **do**
      $v_k^\star \leftarrow \arg\max_v |X_{i,v} \setminus \bigcup_{l=1}^{k-1} T_{i,l}|$
      $T_{i,k} \leftarrow X_{i,v_k^\star} \setminus \bigcup_{l=1}^{k-1} T_{i,l}$ and $\xi_{i,k} \leftarrow \sum_{v \in T_{i,k}} Q_v / |T_{i,k}|.$
   **end for**
   **for** $v \in V \setminus (\bigcup_{k=1}^K T_{i,k})$ **do**
      $k^\star \leftarrow \arg\min_k \|Q_v - \xi_{i,k}\|$
      $T_{i,k^\star} \leftarrow T_{i,k^\star} \cup \{v\}$
   **end for**
   $r_i \leftarrow \sum_{k=1}^K \sum_{v \in T_{i,k}} \|Q_v - \xi_{i,k}\|^2$
**end for**
$i^\star \leftarrow \arg\min_i r_i.$
$S_k \leftarrow T_{i^\star,k}$ for all $k$
**Output:** $(S_k)_{k=1,\dots,K}.$

---

# B    Proofs

In this section, we provide the proofs of the following theorems.

**Theorem 1** *Assume that $\sqrt{\gamma} f(n) = o(1)$. Then under any clustering algorithm $\pi$, the expected proportion of misclassified green nodes tends to 1/2 as $n$ grows large, i.e., $\lim_{n \to \infty} \mathbb{E}[\varepsilon^\pi(n^{(g)})] = 1/2$.*

**Theorem 2** *(i) If there exists a clustering algorithm that classifies the green nodes asymptotically accurately, then we have: $\sqrt{\gamma} f(n) = \omega(1)$.*
*(ii) If there exists an asymptotically accurate clustering algorithm (i.e., classifying all nodes asymptotically accurately), then we have: $\gamma f(n) = \omega(1)$.*

**Theorem 4** *When $\sqrt{\gamma} f(n) = \omega(1)$, Algorithm 1 classifies the green nodes asymptotically accurately.*

**Theorem 5** *When $\gamma f(n) = \omega(1)$, combining Algorithms 1 and 2 yields an asymptotically accurate clustering algorithm.*

**Theorem 6** *Assume that $h(n) = \omega(\frac{\log(n)}{\min\{f(n),n^{1/3}\}})$ and $T = \omega(\frac{n}{\min\{f(n),n^{1/3}\}})$. Then with $M = \Theta(nh(n) + n)$ bits, Algorithm 3, with block size $B = \frac{h(n)n}{\min\{f(n),n^{1/3}\} \log(n)}$ and acquiring the $T$ first*

*columns of A, outputs clusters $\hat{V}_1, \ldots, \hat{V}_K$ such that with high probability, there exists a permutation $\sigma$ of $\{1, \ldots, K\}$ such that:*

$$\frac{1}{n} \left| \bigcup_{1 \leq k \leq K} \hat{V}_k \setminus V_{\sigma(k)} \right| = O\left( \exp(-T \frac{\min\{f(n), n^{1/3}\}}{n}) \right).$$

In the following, we denote by $\lambda_i(X)$ the $i$-th largest singular value of matrix $X$.

## B.1 Proof of Theorem 1

**Preliminaries.** In what follows, we denote by $\sigma^{(g)} \in \{-1, 1\}^{n^{(g)}}$ a vector that represents the repartition of nodes in the two communities, i.e., nodes $v$ and $w$ belong to the same community if and only if $\sigma_v^{(g)} = \sigma_w^{(g)}$. We also denote by $\hat{\sigma}^{(g)} \in \{-1, 1\}^{n^{(g)}}$ the estimate of $\sigma^{(g)}$ that a clustering algorithm could return.

We further introduce the following notation. For any $k > 0$ and any two vectors $x, y \in \{-1, 1\}^k$, we denote by $d_H(x, y) = \sum_{i=1}^{k} 1(x_i \neq y_i)$ the Hamming distance between $x$ and $y$ and define

$$d(x, y) = \frac{1}{k} \min\{d_H(x, y), d_H(x, -y)\}.$$

For an estimate $\hat{\sigma}^{(g)}$ of $\sigma^{(g)}$, the quantity $d(\hat{\sigma}^{(g)}, \sigma^{(g)})$ is exactly the fraction of misclassified green nodes. Hence if estimate $\hat{\sigma}^{(g)}$ is obtained from algorithm $\pi$, we have $\epsilon^\pi(n^{(g)}) = d(\hat{\sigma}^{(g)}, \sigma^{(g)})$. Note that $d(\hat{\sigma}^{(g)}, \sigma^{(g)}) \leq 1/2$.

We first state key lemmas for this proof. Their proofs are postponed to the end of this section.

**Lemma 7** *For any $\alpha < 1/2$ and estimate $\hat{\sigma}^{(g)}$, we have as $n^{(g)} \to \infty$*

$$\mathbb{P}(d(\hat{\sigma}^{(g)}, \sigma^{(g)}) > \alpha) \geq 1 - \frac{n^{(g)} - H(\sigma^{(g)}|A)}{n^{(g)}(1 - H(\alpha))} + o(1),$$

*where $H(\alpha) = -\alpha \log \alpha - (1 - \alpha) \log(1 - \alpha)$ and $H(\sigma^{(g)}|A)$ is the conditional entropy of $\sigma^{(g)}$ knowing $A$.*

**Lemma 8** *As $n^{(g)} \to \infty$, we have:*

$$H(A) - H(A|\sigma^{(g)}) \leq o(n^{(g)}) + O(n^{(g)} \gamma f(n)^2).$$

From the definition of conditional entropy, we have

$$H(\sigma^{(g)}|A) = H(\sigma^{(g)}) - H(A) + H(A|\sigma^{(g)}) = n^{(g)}(1 - o(1)), \tag{1}$$

since $H(\sigma^{(g)}) = \log \binom{n^{(g)}}{n^{(g)}/2} \geq n^{(g)} - \frac{1}{2}\log 2n^{(g)}$ and we have $H(A) - H(A|\sigma^{(g)}) = o(n^{(g)})$ from Lemma 8. As soon as $n^{(g)} \to \infty$, putting (1) into Lemma 7, we see that for any $\alpha < 1/2$ and any estimate $\hat{\sigma}^{(g)}$,

$$\mathbb{P}(d(\hat{\sigma}^{(g)}, \sigma^{(g)}) > \alpha) \to 1.$$

If $\hat{\sigma}^{(g)}$ is a random guess, i.e. for each $v \in V^{(g)}$, $\hat{\sigma}_v^{(g)}$ is equal to 1 or $-1$ with probability $1/2$ independently of the rest, then for any $\alpha < 1/2$, as soon as $n^{(g)} \to \infty$, we have by the weak law of large numbers, $\mathbb{P}(d(\hat{\sigma}^{(g)}, \sigma^{(g)}) > \alpha) \to 1$. Since we have

$$\mathbb{E}[\epsilon^\pi(n^{(g)})] \geq \alpha \mathbb{P}(d(\hat{\sigma}^{(g)}, \sigma^{(g)}) > \alpha),$$

and $\alpha$ can be chosen as close to $1/2$ as desired, the result follows.

## B.2  Proof of Lemma 7

We define the event $E = \{d(\hat{\sigma}^{(g)}, \sigma^{(g)}) > \alpha\}$ and $P_e$ its probability. We have

$$
\begin{aligned}
H(E, \sigma^{(g)}|\hat{\sigma}^{(g)}) &= H(\sigma^{(g)}|\hat{\sigma}^{(g)}) + \underbrace{H(E|\sigma^{(g)}, \hat{\sigma}^{(g)})}_{0} \\
&= H(E|\hat{\sigma}^{(g)}) + H(\sigma^{(g)}|E, \hat{\sigma}^{(g)}) \\
&\leq H(P_e) + P_e \log \binom{n^{(g)}}{n^{(g)}/2} + (1 - P_e)(n^{(g)}H(\alpha) + \log n^{(g)}),
\end{aligned}
$$

where the last inequality follows from $H(E|\hat{\sigma}^{(g)}) \leq H(E) = H(P_e)$ and the fact that

$$|\{\sigma^{(g)}, d(\sigma^{(g)}, \hat{\sigma}^{(g)}) \leq \alpha\}| = \sum_{i=0}^{\alpha n^{(g)}} \binom{n^{(g)}}{i} \leq (n^{(g)}\alpha + 1)\binom{n^{(g)}}{\alpha n^{(g)}} \leq n^{(g)}2^{n^{(g)}H(\alpha)}.$$

Using $H(P_e) \leq 1$ and that $\binom{n^{(g)}}{n^{(g)}/2} \leq 2^{n^{(g)}}$ for sufficiently large $n^{(g)}$, we get

$$P_e \geq \frac{H(\sigma^{(g)}|\hat{\sigma}^{(g)}) - 1 - n^{(g)}H(\alpha) - \log n^{(g)}}{n^{(g)}(1 - H(\alpha)) - \log n^{(g)}}.$$

The claim follows from the data processing inequality which ensures $H(\sigma^{(g)}|\hat{\sigma}^{(g)}) \geq H(\sigma^{(g)}|A)$.

## B.3  Proof of Lemma 8

Thanks to independence, we have

$$H(A) - H(A|\sigma^{(g)}) = H(A^{(g)}) - H(A^{(g)}|\sigma^{(g)}) + H(A^{(r)}) - H(A^{(r)}|\sigma^{(g)})$$

We first deal with the first term $H(A^{(g)}) - H(A^{(g)}|\sigma^{(g)})$. By the concavity of $p \mapsto H(p)$, we have $H(A^{(g)}) \leq \binom{n^{(g)}}{2} H\left(\frac{p+q}{2}\right)$ and $H(A^{(g)}|\sigma^{(g)}) = 2\binom{n^{(g)}/2}{2} H(p) + \left(\frac{n^{(g)}}{2}\right)^2 H(q)$. Hence, we get

$$
\begin{aligned}
H(A^{(g)}) - H(A^{(g)}|\sigma^{(g)}) &\leq \binom{n^{(g)}}{2} H(\frac{p+q}{2}) - 2\binom{n^{(g)}/2}{2} H(p) - (\frac{n^{(g)}}{2})^2 H(q) \\
&= \frac{(n^{(g)})^2}{4} \left( 2H(\frac{p+q}{2}) - H(p) - H(q) \right) + o(n^{(g)}) \\
&= \frac{(n^{(g)})^2}{4} \left( p \log \frac{2p}{p+q} + q \log \frac{2q}{p+q} \right) \\
&\quad + \frac{(n^{(g)})^2}{4} \left( (1-p) \log \frac{2-2p}{2-p-q} + (1-q) \log \frac{2-2q}{2-p-q} \right) + o(n^{(g)}) \\
&\leq \frac{(n^{(g)})^2}{4} \left( \frac{(p-q)^2}{p+q} + \frac{(p-q)^2}{2-p-q} \right) + o(n^{(g)}) \\
&= o(n^{(g)}) + o(n^{(g)} \gamma f(n)).
\end{aligned}
$$

We denote by $A_v^{(r)}$ the row vector of $A^{(r)}$ corresponding to $v \in V^{(r)}$. For the second term, by independence we have

$$
H(A^{(r)}) - H(A^{(r)}|\sigma^{(g)}) = (n - n^{(g)})\left(H(A_v^{(r)}) - H(A_v^{(r)}|\sigma^{(g)})\right).
$$

For a vector $x \in \{-1, 1\}^{V^{(g)}}$ and $\sigma^{(g)}$, we define $|x|^+ = \sum_{v \in V^{(g)},\, \sigma_v^{(g)}=1} x_v$, $|x|^- = \sum_{v \in V^{(g)},\, \sigma_v^{(g)}=-1} x_v$ and $|x| = |x|^+ + |x|^-$. For a given $\sigma^{(g)}$, we have

$$
\mathbb{P}[A_v^{(r)} = x|\sigma^{(g)}] = \zeta(|x|^+, |x|^-),
$$

where

$$
\zeta(i, j) = \frac{((\frac{p}{1-p})^i (\frac{q}{1-q})^j + (\frac{p}{1-p})^j (\frac{q}{1-q})^i)(1-p)^{\frac{m}{2}}(1-q)^{\frac{m}{2}}}{2}.
$$

Since $\sigma^{(g)}$ is uniformly distributed,

$$
\begin{aligned}
\mathbb{P}[A_v^{(g)} = x] &= \binom{n^{(g)}}{n^{(g)}/2}^{-1} \sum_{\sigma^{(g)}:\sum_{v \in V^{(g)}} \sigma_v^{(g)}=0} \mathbb{P}[A_v^{(g)} = x|\sigma^{(g)}] \\
&= \binom{n^{(g)}}{n^{(g)}/2}^{-1} \sum_{i=0}^{|x|} \binom{|x|}{i} \binom{n^{(g)} - |x|}{n^{(g)}/2 - i} \zeta(i, |x| - i) = \eta(|x|),
\end{aligned}
$$

where

$$
\eta(k) = \frac{\sum_{i=0}^{k} \binom{n^{(g)}/2}{i} \binom{n^{(g)}/2}{k-i} \zeta(i, k-i)}{\sum_{i=0}^{k} \binom{n^{(g)}/2}{i} \binom{n^{(g)}/2}{k-i}}.
$$

Since $\eta(0) = \zeta(0,0)$, $\eta(1) = \zeta(1,0) = \zeta(0,1)$, and $\binom{n^{(g)}}{k}\eta(k) = \sum_{i=0}^{k}\binom{n^{(g)}/2}{i}\binom{n^{(g)}/2}{k-i}\zeta(i,k-i)$,

$$
H(A_v^{(r)}) - H(A_v^{(r)}|\sigma^{(g)})
$$

$$
= -\sum_{k=0}^{n^{(g)}}\binom{n^{(g)}}{k}\eta(k)\log\eta(k) + \sum_{i=0}^{n^{(g)}/2}\sum_{j=0}^{n^{(g)}/2}\binom{n^{(g)}/2}{i}\binom{n^{(g)}/2}{j}\zeta(i,j)\log\zeta(i,j)
$$

$$
= \sum_{i=0}^{n^{(g)}/2}\sum_{j=0}^{n^{(g)}/2}\binom{n^{(g)}/2}{i}\binom{n^{(g)}/2}{j}\zeta(i,j)\log\frac{\zeta(i,j)}{\eta(i+j)}
$$

$$
= \sum_{k=2}^{n^{(g)}}\sum_{i=0}^{k}1_{i\leq n^{(g)}/2}1_{k-i\leq n^{(g)}/2}\binom{n^{(g)}/2}{i}\binom{n^{(g)}/2}{k-i}\zeta(i,k-i)\log\frac{\zeta(i,k-i)}{\eta(k)}
$$

$$
\leq \sum_{k=2}^{n^{(g)}}\sum_{i=0}^{k}1_{i\leq n^{(g)}/2}1_{k-i\leq n^{(g)}/2}\binom{n^{(g)}/2}{i}\binom{n^{(g)}/2}{k-i}\zeta(i,k-i)k\log(\frac{p}{q})
$$

$$
\leq \sum_{2\leq k\leq n^{(g)}}(n^{(g)}p)^k k\log(\frac{a}{b})
$$

$$
= O((n^{(g)})^2 p^2),
$$

where the last equality stems from $n^{(g)}p = o(1)$. Thus,

$$
(n-m)\big(H(A_v^{(r)}) - H(A_v^{(r)}|\sigma^{(g)})\big) = O(n(n^{(g)})^2 p^2) = O(n^{(g)}\frac{n^{(g)}}{n}f(n)^2) = O(n^{(g)}\gamma f(n)^2),
$$

and the lemma follows since $f(n) \geq 1$ so that $f(n)^2 \geq f(n)$.

## B.4 Proof of Theorem 2

In the remaining proofs, we use $m$ instead of $n^{(g)}$ to denote the number of green nodes. We first consider case (i) with $\gamma = \Theta(1)$. In this case, a necessary condition for the existence of an asymptotically accurate clustering algorithm is that the fraction of green nodes outside the largest connected component of the observed graph vanishes as $n \to \infty$. This condition imposes that $f(n) \to \infty$.

We now consider case (i) with $m = o(n)$, i.e. $\gamma = o(1)$. Denote by $\Phi$ the true hidden partition $(V_1^{(g)}, V_2^{(g)})$ for green nodes. Let $\mathbb{P}_\Phi$ be the probability measure capturing the randomness in the observations assuming that the network structure is described by $\Phi$. We also introduce a slightly different structure $\Psi$. The latter is described by clusters $V_1'^{(g)} = V_1^{(g)} \cup \{v_2\} \setminus \{v_1\}$, $V_2'^{(g)} = V_2^{(g)} \cup \{v_1\} \setminus \{v_2\}$ with arbitrary selected $v_1 \in V_1^{(g)}$ and $v_2 \in V_2^{(g)}$.

Let $\pi \in \Pi$ denote a clustering algorithm for green nodes with output $(\hat{V}_1^{(g)}, \hat{V}_2^{(g)})$, and let $\mathcal{E} = \hat{V}_1^{(g)} \triangle V_1^{(g)}$ be the set of misclassified nodes under $\pi$. Note that in general in our proofs, we

always assume without loss of generality that $|\hat{V}_1^{(g)} \triangle V_1^{(g)}| \leq |\hat{V}_1^{(g)} \triangle V_2^{(g)}|$, so that the set of misclassified nodes is really $\mathcal{E}$. Further define $\mathcal{B} = \{v_1 \in \hat{V}_1^{(g)}\}$ as the set of events where node $v_1$ is correctly classified. We have $\varepsilon(m) = |\mathcal{E}|/m$.

Let $x_{i,j}$ be equal to one if there is an edge between nodes $i$ and $j$ and zero otherwise. With a slight abuse of notation, we define the boolean functions $p(\cdot)$ and $q(\cdot)$ as follows: $p(1) = af(n)/n = p$, $q(1) = bf(n)/n = q$ and $p(0) = 1 - p(1)$, $q(0) = 1 - q(1)$. We introduce $L$ (a quantity that resembles the log-likelihood ratio between $\mathbb{P}_\Phi$ and $\mathbb{P}_\Psi$) as:

$$L = \sum_{i \in V_1'^{(g)}} \log \frac{q(x_{i,v_1})p(x_{i,v_2})}{p(x_{i,v_1})q(x_{i,v_2})} + \sum_{i \in V_2'^{(g)}} \log \frac{p(x_{i,v_1})q(x_{i,v_2})}{q(x_{i,v_1})p(x_{i,v_2})}$$
$$+ \sum_{v \in V^{(r)}} \log \frac{\prod_{i \in V_1'^{(g)}} p(x_{v,i}) \prod_{i \in V_2'^{(g)}} q(x_{v,i}) + \prod_{i \in V_1'^{(g)}} q(x_{v,i}) \prod_{i \in V_2'^{(g)}} p(x_{v,i})}{\prod_{i \in V_1^{(g)}} p(x_{v,i}) \prod_{i \in V_2^{(g)}} q(x_{v,i}) + \prod_{i \in V_1^{(g)}} q(x_{v,i}) \prod_{i \in V_2^{(g)}} p(x_{v,i})},$$

In what follows, we establish a relationship between $\mathbb{E}[\varepsilon(m)]$ and $L$. For any function $g(m)$,

$$\mathbb{P}_\Psi\{L \leq g(m)\} \quad = \quad \mathbb{P}_\Psi\{L \leq g(m), \bar{\mathcal{B}}\} + \mathbb{P}_\Psi\{L \leq g(m), \mathcal{B}\}. \tag{2}$$

We have:

$$\begin{aligned} \mathbb{P}_\Psi\{L \leq g(m), \bar{\mathcal{B}}\} \quad &= \quad \int_{\{L \leq g(m), \bar{\mathcal{B}}\}} d\mathbb{P}_\Psi \\ &= \quad \int_{\{L \leq g(m), \bar{\mathcal{B}}\}} \prod_{i \in V_1'} \frac{\nu(x_{i,v_1})\nu(x_{i,v_2})}{p(x_{i,v_1})q(x_{i,v_2})} \prod_{i \in V_2'} \frac{\nu(x_{i,v_1})\nu(x_{i,v_2})}{q(x_{i,v_1})p(x_{i,v_2})} d\mathbb{P}_\Phi \\ &\leq \quad \exp(g(m))\mathbb{P}_\Phi\{L \leq g(m), \bar{\mathcal{B}}\} \leq \exp(g(m))\mathbb{P}_\Phi\{\bar{\mathcal{B}}\} \\ &\leq \quad 2\exp(g(m))\mathbb{E}_\Phi[\varepsilon(m)], \end{aligned} \tag{3}$$

where the last inequality comes from the fact that,

$$\mathbb{P}_\Phi\{\mathcal{B}\} \quad \geq \quad 1 - \mathbb{P}_\Phi\{v_1 \notin \hat{V}_1^{(g)}\} \geq 1 - 2\mathbb{E}_\Phi[\varepsilon(n)].$$

We also have:

$$\mathbb{P}_\Psi\{L \leq g(m), \mathcal{B}\} \leq \mathbb{P}_\Psi\{\mathcal{B}\} \leq 2\mathbb{E}_\Psi[\varepsilon(m)]. \tag{4}$$

By (3) and (4)

$$\mathbb{P}_\Psi\{L \leq g(n)\} \leq 2\mathbb{E}_\Phi[\varepsilon(n)]\exp(g(n)) + 2\mathbb{E}_\Psi[\varepsilon(n)].$$

Since $\mathbb{E}_\Phi[\varepsilon(n)] = \mathbb{E}_\Psi[\varepsilon(n)] = \mathbb{E}[\varepsilon(n)]$ and $\mathbb{E}[\varepsilon(n)] = o(1)$, choosing $g(m) = \log\left(\frac{1}{8\mathbb{E}[\varepsilon(m)]}\right)$, we obtain:

$$\lim_{m \to \infty} \inf \mathbb{P}_\Psi\{L \geq \log\left(\frac{1}{8\mathbb{E}[\varepsilon(m)]}\right)\} > \frac{1}{2}. \tag{5}$$

By Chebyshev's inequality, $\mathbb{P}_\Psi\{L \geq \mathbb{E}_\Psi[L] + 2\sigma_\Psi[L]\} \leq \frac{1}{4}$. Therefore, to be valid the above inequality,

$$\mathbb{E}_\Psi[L] + 2\sigma_\Psi[L] \geq \log\left(\frac{1}{8\mathbb{E}[\varepsilon(m)]}\right), \tag{6}$$

which implies that $\mathbb{E}_\Psi[L] + 2\sigma_\Psi[L] = \omega(1)$ since $\mathbb{E}[\varepsilon(m)] = o(1)$.

We define $KL(p,q) = p\log(p/q) + (1-p)\log((1-p)/(1-q))$. From the definition of $L$, we can easily bound $\mathbb{E}_\Psi[L]$ and $\sigma_\Psi[L]^2$ :

$$
\begin{aligned}
\mathbb{E}_\Psi[L] \leq & m \cdot (KL(p,q) + KL(q,p)) \\
& + n \sum_{0 \leq i,j \leq \frac{m}{2}-1} \binom{m/2-1}{i}\binom{m/2-1}{j} \frac{p^{i+1}q^j + p^j q^{i+1}}{2} \log\frac{p^{i+1}q^j + p^j q^{i+1}}{p^i q^{j+1} + p^{j+1}q^i} \\
\leq & m \cdot (KL(p,q) + KL(q,p)) + n \sum_{1 \leq k \leq m} m^k p^{k+1} k \log\frac{p}{q} \\
\leq & O(\gamma f(n)) + np\log\frac{a}{b} \sum_{1 \leq k \leq m} km^k p^k \leq O(\gamma f(n)) + np\log\frac{a}{b}\sum_{k=1}^{\infty} k(mp)^k \\
\leq & O(\gamma f(n)) + np\log\frac{a}{b}\sum_{k=1}^{\infty}(2mp)^k
\end{aligned}
$$

$$
\begin{aligned}
\sigma_\Psi[L]^2 \leq & m\left((p+q)(\log\frac{a}{b})^2 + (2-p-q)(\log\frac{1-q}{1-p})^2\right) \\
& + n \sum_{0 \leq i,j \leq \frac{m}{2}-1} \binom{m/2-1}{i}\binom{m/2-1}{j} \frac{p^{i+1}q^j + p^j q^{i+1}}{2} \left(\log\frac{p^{i+1}q^j + p^j q^{i+1}}{p^i q^{j+1} + p^{j+1}q^i}\right)^2 \\
\leq & 4mp(\log\frac{a}{b})^2 + n \sum_{1 \leq k \leq m} m^k p^{k+1} k^2 (\log\frac{a}{b})^2 \\
\leq & O(\gamma f(n)) + np(\log\frac{a}{b})^2 \sum_{k=1}^{\infty} k^2(mp)^k \leq O(\gamma f(n)) + np(\log\frac{a}{b})^2 \sum_{k=1}^{\infty}(3mp)^k. \tag{7}
\end{aligned}
$$

Therefore, the necessary condition for $\mathbb{E}_\Psi[L] + 2\sigma_\Psi[L] = \omega(1)$ is that $np\sum_{k=1}^{\infty}(3mp)^k = \omega(1)$. We conclude this proof from that $np\sum_{k=1}^{\infty}(3mp)^k = \omega(1)$ if and only if $\gamma f(n)^2 = \omega(1)$.

We now prove point (ii). Note that the probability for a red node to be isolated is at least $(1-af(n)/n)^{\gamma n} \approx \exp(-a\gamma f(n))$. If there exists an asymptotically accurate clustering algorithm, then the fraction of such isolated red nodes should vanishes and hence $\gamma f(n) \to \infty$.

## B.5 Proof of Theorem 4

The proof proceeds in two steps. Step 1. We first establish that if $\frac{\sigma_K^{(g)}}{\sqrt{m\hat{p}^{(g)}}} \cdot 1_{\{m\hat{p}^{(g)} \geq 50\}} = \omega(1)$, then the spectral method applied to the matrix $A^{(g)}$ is asymptotically accurate. We also show that

if $\frac{\sigma'_K}{\sqrt{m\hat{p}'}} \cdot 1_{\{m\hat{p}' \geq 50\}} = \omega(1)$, then the spectral method applied to the matrix of indirect edges $A'$ is asymptotically accurate. Step 2. We show that if $\gamma f(n) = \omega(1)$, then $\frac{\sigma_K^{(g)}}{\sqrt{m\hat{p}^{(g)}}} \cdot 1_{\{m\hat{p}^{(g)} \geq 50\}} = \omega(1)$ with high probability (w.h.p.), and if $\gamma f(n) = O(1)$ and $\sqrt{\gamma} f(n) = \omega(1)$, then $\frac{\sigma'_K}{\sqrt{m\hat{p}'}} \cdot 1_{\{m\hat{p}' \geq 50\}} = \omega(1)$ w.h.p..

**Preliminaries.** We first state three lemmas to analyze the performance of Approx, PowerMethod, and Detection algorithms. Their proofs are postponed to the end of this section. In what follows, let $V = \{1, \ldots, n\}$ and let $A \in \mathbb{R}^{n \times n}$. For any matrix $Z \in \mathbb{R}^{n \times n}$, $\lambda_K(Z)$ denotes the $K$-th largest singular value of $Z$.

**Lemma 9** *With probability $1 - O(1/n)$, the output $(Q, \sigma_K)$ of the PowerMethod algorithm with input $(A, V, K)$ (Algorithm 5) satisfies that $\sigma_K = \Theta(\lambda_K(A))$.*

**Lemma 10** *Let $A, M \in \mathbb{R}^{n \times n}$ and let $M = U \Lambda U^T$ be the SVD of $M$ where $\Lambda \in \mathbb{R}^{K \times K}$. Assume that $\|A - M\| = o(\lambda_K(M))$, the output $(Q, \sigma_K)$ of the PowerMethod algorithm (Algorithm 5) with input $(A, V, K)$ satisfies:*

$$\|U_\perp^T Q\| = O\left(\frac{\|A - M\|}{\lambda_K(M)}\right) = o(1),$$

*where $U_\perp$ is an orthonormal basis of the space perpendicular to the linear span of $U$.*

**Lemma 11** *Assume that the set $V$ is partitioned into $K$ subsets $(V_k)_{1 \leq k \leq K}$. Further assume that for any $k$, $\frac{|V_k|}{n} > 0$ does not depend on $n$. Let $W$ be the $V \times K$ matrix with for all $(v, k)$, $W_{vk} = 1/\sqrt{|V_k|}$ if $v \in V_k$ and $0$ otherwise. Let $W_\perp$ be an orthonormal basis of the space perpendicular to the linear span of $W$. The output $(S_k)_{1 \leq k \leq K}$ of the Detection algorithm with input $(Q, V, K)$ satisfies: if $\|W_\perp^T Q\| = o(1)$, then there exists a permutation $\zeta$ of $\{1, \ldots, K\}$ such that*

$$\frac{\left|\bigcup_{k=1}^K S_k \setminus V_{\zeta(k)}\right|}{n} = O\left(\|W_\perp^T Q\|^2\right).$$

**Step 1.** We use the notations introduced in the pseudo-codes of the various algorithms. Let $M^{(g)} = \mathbb{E}[A^{(g)}]$ and $M' = \mathbb{E}[A']$. Let $A_\Gamma^{(g)} = (A_{vw}^{(g)})_{v,w \in \Gamma^{(g)}}$ and $M_\Gamma^{(g)} = (M_{vw}^{(g)})_{v,w \in \Gamma^{(g)}}$. Analogously, we define $A'_\Gamma = (A'_{vw})_{v,w \in \Gamma'}$ and $M'_\Gamma = (M'_{vw})_{v,w \in \Gamma'}$.

We prove that if $\frac{\sigma'_K}{\sqrt{m\hat{p}'}} \cdot 1_{\{m\hat{p}' \geq 50\}} = \omega(1)$, then the spectral method applied to the matrix of indirect edges $A'$ is asymptotically accurate. We omit the proof of the asymptotic accuracy of the spectral method applied to $A^{(g)}$ under the condition $\frac{\sigma_K^{(g)}}{\sqrt{m\hat{p}^{(g)}}} \cdot 1_{\{m\hat{p}^{(g)} \geq 50\}} = \omega(1)$ (since it can be conducted in the same way).

Recall that $\sigma'_K$ denotes the $K$-th largest singular value of the trimmed matrix $A'_\Gamma$. Observe that by assumption, for $n$ large enough, $m\hat{p}' \geq 50$. Hence applying the law of large numbers, we can conclude that the largest singular value $\xi_1$ of $A'$ scales at most as $m\hat{p}'$ w.h.p.. Since $\sigma'_K \leq \sigma'_1 \leq \xi_1$ (where $\sigma'_1$ is the largest singular value of $A'_\Gamma$) and $\frac{\sigma'_K}{\sqrt{m\hat{p}'}} = \omega(1)$, we deduce that $m\hat{p}' = \omega(1)$ w.h.p.. Hence the trimming step in the Approx algorithm applied to $(A', \hat{p}', V^{(g)}, K)$ does remove a negligible proportion of green nodes, i.e., w.h.p. $|V^{(g)} \setminus \Gamma| = o(|V^{(g)}|)$ or equivalently $|\Gamma'| = m(1 + o(1))$.

Observe that w.h.p., $\hat{p}' = \frac{\sum_{u,v} M'_{uv}}{m^2}(1 + o(1)) = \Theta(\max_{uv}\{M'_{uv}\})(1 + o(1))$ by the law of large numbers and $\sum_{w \in \Gamma'} A'_{vw} = O(m\hat{p}')$ for all $v \in \Gamma'$. From random matrix theory [2], with probability $1 - O(1/m)$, $\|A'_\Gamma - M'_\Gamma\| = O(\sqrt{m\hat{p}'})$. Next we apply Lemma 9 to $(A'_\Gamma, \Gamma', K)$ and deduce that $\sigma'_K = \Theta(\lambda_K(A'_\Gamma))$ w.h.p.. From $\lambda_K(M'_\Gamma) \geq \lambda_K(A'_\Gamma) - \|A'_\Gamma - M'_\Gamma\|$, and $\frac{\sigma'_K}{\sqrt{m\hat{p}'}} = \omega(1)$, we deduce that w.h.p.,

$$\frac{\lambda_K(M'_\Gamma)}{\|A'_\Gamma - M'_\Gamma\|} = \omega(1).$$

If $M'_\Gamma = U \Lambda U^T$, we deduce from Lemma 10 applied to $A'_\Gamma$ and $M'_\Gamma$ that w.h.p., $\|U^T_\perp Q\| = o(1)$. We can now apply Lemma 11 replacing $V$ by $\Gamma'$ and $V_k$ by $\Gamma' \cap V_k^{(g)}$. Observe that the linear span of $U$ coincides with that of $W$ (refer to Lemma 11 for the definition of $W$). Hence, w.h.p., the nodes $\Gamma'$ are accurately classified, and so are the nodes in $V^{(g)}$.

**Step 2.** We distinguish two cases: 1. $\gamma f(n) = \omega(1)$; 2. $\gamma f(n) = O(1)$ and $\sqrt{\gamma} f(n) = \omega(1)$.

**Case 1.** Assume that $\gamma f(n) = \omega(1)$. By the law of large numbers, $m\hat{p}^{(g)} = \Theta(\gamma f(n))$ w.h.p.. Since $\lambda_K(M^{(g)}_\Gamma) = \Omega(\gamma f(n))$, $\|A^{(g)}_\Gamma - M^{(g)}_\Gamma\| = \Theta(\sqrt{m\hat{p}^{(g)}}) = \Theta(\sqrt{\gamma f(n)})$ and $\lambda_K(A^{(g)}_\Gamma) \geq \lambda_K(M^{(g)}_\Gamma) - \|A^{(g)}_\Gamma - M^{(g)}_\Gamma\|$, we get $\frac{\lambda_K(A^{(g)}_\Gamma)}{\sqrt{m\hat{p}^{(g)}}} = \omega(1)$ w.h.p.. Since $\sigma^{(g)}_K = \Theta(\lambda_K(A^{(g)}_\Gamma))$ from Lemma 9, w.h.p.

$$\frac{\sigma^{(g)}_K}{\sqrt{m\hat{p}^{(g)}}} \cdot 1_{\{m\hat{p}^{(g)} \geq 50\}} = \omega(1).$$

**Case 2.** Assume that $\gamma f(n) = O(1)$ and $\sqrt{\gamma} f(n) = \omega(1)$. We first compute $M'_{ij}$ for any $i, j \in V^{(g)}$. For notational simplicity, $\alpha_k = \frac{|V_k^{(g)}|}{n}$ and $\beta_k = \frac{|V_k^{(r)}|}{n}$.

(i) Let $i, j$ be two green nodes belonging to the same community, i.e., $i, j \in V_k^{(g)}$. Let $v \in V_k^{(r)}$. We have:

$$\mathbb{P}\left[A_{vi} = 1 = A_{vj}, \sum_{w \in V^{(g)}} A_{vw} = 2\right] = p^2(1-p)^{\alpha_k n - 2} \prod_{l \neq k}(1-q)^{\alpha_l n}.$$

This probability is equivalent to $p^2 \exp(-\alpha_k p n - \sum_{l \neq k} \alpha_l q n)$ when $n \to \infty$. Similarly, when $v \in V_{k'}^{(g)}$ for some $k' \neq k$, the probability $\mathbb{P}[A_{vi} = 1 = A_{vj}, \sum_{w \in V^{(g)}} A_{vw} = 2]$ is equivalent to

$q^2 \exp(-\alpha_{k'} pn - \sum_{l \neq k'} \alpha_l qn)$ when $n \to \infty$. We deduce that:

$$M'_{ij} \sim p^2 n \beta_k \eta_k + q^2 n \sum_{k' \neq k} \beta_{k'} \eta_{k'}, \quad \text{as } n \to \infty, \tag{8}$$

where $\eta_k = \exp(-\alpha_k pn - \sum_{l \neq k} \alpha_l qn)$.

(ii) Let $i, j$ be two green nodes belonging to different communities, i.e., $i \in V_k^{(g)}$ and $j \in V_\ell^{(g)}$, for $k \neq \ell$. Using the same analysis as above, we have:

$$M'_{ij} \sim pq(\beta_k \eta_k + \beta_\ell \eta_\ell) n + q^2 n \sum_{k' \notin \{k, \ell\}} \beta_{k'} \eta_{k'} \quad \text{as } n \to \infty. \tag{9}$$

From (8)-(9) and the law of large numbers, we get w.h.p., $m\hat{p}' = \Theta(\gamma f(n)^2)$ (this comes from the facts that $\gamma f(n) = O(1)$ and $\alpha_k pn = \Theta(\gamma pn) = \Theta(\gamma f(n))$). As a consequence, $m\hat{p}' = \omega(1)$ w.h.p.. Thus, in the trimming process in the Approx algorithm applied to $(A', \hat{p}', V^{(g)}, K)$, we must have $|V^{(g)} \setminus \Gamma'| = o(|V^{(g)}|)$ w.h.p..

We also deduce from the above analysis that we can represent $M'_\Gamma$ as follows:

$$M'_\Gamma = M^{(g)}_{\Gamma', K} \Lambda' (M^{(g)}_{\Gamma', K})^T,$$

where $M^{(g)}_{\Gamma', K}$ is a $\Gamma' \times K$ matrix where the $k$-th column of $M^{(g)}_{\Gamma', K}$ is the column vector of $M^{(g)}_\Gamma$ corresponding to $v \in V_k^{(g)}$, and $\Lambda'$ is a $K \times K$ diagonal matrix where $k$-th element is $\beta_k \eta_k n$. Since $\frac{\|M^{(g)}_{\Gamma', K} \boldsymbol{x}\|}{\|\boldsymbol{x}\|} = \Omega(\sqrt{mp})$ for any $\boldsymbol{x} \in \mathbb{R}^{K \times 1}$, $\lambda_K(M'_\Gamma) = \Omega(mnp^2 \min_{1 \leq k \leq K} \eta_k) = \Omega(\gamma f(n)^2)$. By the law of large numbers, w.h.p., $m\hat{p}' = \Theta(\gamma f(n)^2)$. Then, as in the analysis of Case 1, we conclude that w.h.p.

$$\frac{\sigma'_K}{\sqrt{m\hat{p}'}} \cdot 1_{\{m\hat{p}' \geq 50\}} = \omega(1).$$

## B.6 Proof of Lemma 9

To conclude $\sigma_K = \Theta(\lambda_K(A))$, we show that $\sigma_K = O(\lambda_K(A))$ (Step 1) and $\sigma_K = \Omega(\lambda_K(A))$ (Step 2).

**Step 1.** When $\|A\| = \Theta(\lambda_K(A))$, this is trivial since singular values of $R_{\tau^\star}$ have to be less than $\|A\|$. Let $\|A\| = \omega(\lambda_K(A))$. Then, there exists $\ell < K$ such that $\lambda_\ell(A) = \omega(\lambda_K(A))$ and $\lambda_{\ell+1}(A) = \Theta(\lambda_K(A))$. We denote by $\tilde{U}_j \tilde{\Lambda} \tilde{U}_j^T$ be the SVD of rank $j$ approximation of $A$. Let $Q_{K, \tau^\star}$ denote the $K$-th column vector of $Q_{\tau^\star}$. Analogously with Step 1 of the proof of Lemma 10, we can show that $\|\tilde{U}_j^T Q_{K, \tau^\star}\| = O(\frac{\lambda_K(A)}{\lambda_j(A)})$ for all $j \leq \ell$. Therefore, $\sigma_k = O(\lambda_K(A))$.

**Step 2.** When $\lambda_n(A) = \Theta(\lambda_K(A))$, this is trivial since singular values of $R_{\tau^\star}$ have to be larger than $\lambda_n(A)$. Let $\lambda_n(A) = o(\lambda_K(A))$. Then, there exists $\ell \geq K$ such that $\lambda_{\ell+1}(A) = o(\lambda_K(A))$

and $\lambda_\ell(A) = \Theta(\lambda_K(A))$. Analogously with Step 1 of the proof of Lemma 10, we can show that $\|(\tilde{U}_\ell)_\perp^T Q_{K,\tau^*}\| = O(\frac{\lambda_{\ell+1}(A)}{\lambda_\ell(A)}) = o(1)$, where $(\tilde{U}_\ell)_\perp$ is an orthonormal basis of the perpendicular to the linear span of $\tilde{U}_\ell$. Therefore, $\sigma_k = \Omega(\lambda_\ell(A)) = \Omega(\lambda_K(A))$.

## B.7 Proof of Lemma 10

We denote by $\tilde{A} = \tilde{U}\tilde{\Lambda}\tilde{U}^T$ be the SVD of rank $K$ approximation of $A$. Let $U_\perp$ and $\tilde{U}_\perp$ be orthonormal bases of the perpendicular spaces to the linear spans of $U$ and $\tilde{U}$, respectively. Since

$$\|U_\perp^T Q_{\tau^*}\| = \|U_\perp^T(\tilde{U}\tilde{U}^T + \tilde{U}_\perp\tilde{U}_\perp^T)Q_{\tau^*}\| \leq \|U_\perp^T\tilde{U}\tilde{U}^T Q_{\tau^*}\| + \|U_\perp^T\tilde{U}_\perp\tilde{U}_\perp^T Q_{\tau^*}\|$$
$$\leq \|U_\perp^T\tilde{U}\|\|\tilde{U}^T Q_{\tau^*}\| + \|U_\perp^T\tilde{U}_\perp\|\|\tilde{U}_\perp^T Q_{\tau^*}\| \leq \|U_\perp^T\tilde{U}\| + \|\tilde{U}_\perp^T Q_{\tau^*}\|,$$

to conclude this proof, we will show that $\|\tilde{U}_\perp^T Q_{\tau^*}\| = O\left(\frac{\|A-M\|}{\lambda_K(M)}\right)$ and $\|U_\perp^T\tilde{U}\| = O\left(\frac{\|A-M\|}{\lambda_K(M)}\right)$.

**Step 1.** $\|\tilde{U}_\perp^T Q_{\tau^*}\| = O\left(\frac{\|A-M\|}{\lambda_K(M)}\right)$: Let $\boldsymbol{x}_1$ be the right singular vector of $\tilde{U}_\perp^T Q_{\tau+1}$ corresponding to the largest singular value and $\tilde{\boldsymbol{x}}_1$ be a $K \times 1$ vector such that $\boldsymbol{x}_1 = R_{\tau+1}\tilde{\boldsymbol{x}}_1$. Then,

$$\|\tilde{U}_\perp^T Q_{\tau+1}\|_2^2 = \frac{\|\tilde{U}_\perp^T Q_{\tau+1}\boldsymbol{x}_1\|_2^2}{\|\boldsymbol{x}_1\|_2^2} = \frac{\|\tilde{U}_\perp^T Q_{\tau+1}R_{\tau+1}\tilde{\boldsymbol{x}}_1\|_2^2}{\|R_{\tau+1}\tilde{\boldsymbol{x}}_1\|_2^2}$$
$$= \frac{\|\tilde{U}_\perp^T Q_{\tau+1}R_{\tau+1}\tilde{\boldsymbol{x}}_1\|_2^2}{\|\tilde{U}^T Q_{\tau+1}R_{\tau+1}\tilde{\boldsymbol{x}}_1\|_2^2 + \|\tilde{U}_\perp^T Q_{\tau+1}R_{\tau+1}\tilde{\boldsymbol{x}}_1\|_2^2}$$
$$= \frac{\|\tilde{U}_\perp^T A Q_\tau\tilde{\boldsymbol{x}}_1\|_2^2}{\|\tilde{U}^T A Q_\tau\tilde{\boldsymbol{x}}_1\|_2^2 + \|\tilde{U}_\perp^T A Q_\tau\tilde{\boldsymbol{x}}_1\|_2^2}$$
$$\leq \frac{\|A-M\|_2^2}{(\lambda_K(M) - \|A-M\|_2)^2(1 - \|\tilde{U}_\perp^T Q_\tau\|_2^2) + \|A-M\|_2^2}, \tag{10}$$

where the last inequality stems from that

$$\|\tilde{U}_\perp^T A Q_\tau\tilde{\boldsymbol{x}}_1\|_2 \leq \|\tilde{U}_\perp^T A\|\|Q_\tau\|\|\tilde{\boldsymbol{x}}_1\|_2 \leq \|\tilde{U}_\perp^T A\|\|\tilde{\boldsymbol{x}}_1\|_2 = \lambda_{K+1}(A)\|\tilde{\boldsymbol{x}}_1\|_2 \leq \|(A-M)\|\|\tilde{\boldsymbol{x}}_1\|_2$$
$$\|\tilde{U}^T A Q_\tau\tilde{\boldsymbol{x}}_1\|_2 = \|(\tilde{U}^T A \tilde{U}\tilde{U}^T Q_\tau + \tilde{U}^T A \tilde{U}_\perp\tilde{U}_\perp^T Q_\tau)\tilde{\boldsymbol{x}}_1\|_2 = \|\tilde{U}^T A \tilde{U}\tilde{U}^T Q_\tau\tilde{\boldsymbol{x}}_1\|_2$$
$$\geq \lambda_K(A)\|\tilde{U}^T Q_\tau\tilde{\boldsymbol{x}}_1\|_2 \geq (\lambda_K(M) - \|A-M\|)\|\tilde{U}^T Q_\tau\tilde{\boldsymbol{x}}_1\|$$
$$\geq (\lambda_K(M) - \|A-M\|)\sqrt{1 - \|\tilde{U}_\perp^T Q_\tau\|_2^2}\|\tilde{\boldsymbol{x}}_1\|.$$

Let $\zeta = \frac{\|A-M\|_2^2}{(\lambda_K(M) - \|A-M\|_2)^2}$. Since $\frac{\|A-M\|_2}{\lambda_K(M)} = o(1)$, $\zeta = O(\frac{\|A-M\|_2^2}{\lambda_K(M)^2}) = o(1)$. Then, from (10),

$$1 - \|\tilde{U}_\perp^T Q_{\tau+1}\|_2^2 \geq 1 - \frac{\zeta}{1 - \|\tilde{U}_\perp^T Q_\tau\|_2^2 + \zeta} = \frac{1 - \|\tilde{U}_\perp^T Q_\tau\|_2^2}{1 - \|\tilde{U}_\perp^T Q_\tau\|_2^2 + \zeta}.$$

When $1 - \|\tilde{U}_\perp^T Q_\tau\|_2^2 \le \zeta$, $1 - \|\tilde{U}_\perp^T Q_{\tau+1}\|_2^2 \ge \frac{1-\|\tilde{U}_\perp^T Q_\tau\|_2^2}{2\zeta}$. From this, one can easily check that when $\tau \ge \frac{\log(\zeta/(1-\|\tilde{U}_\perp^T Q_0\|_2^2))}{\log(1/2\zeta)}$, $1 - \|\tilde{U}_\perp^T Q_\tau\|_2^2 \ge \zeta$, $1 - \|\tilde{U}_\perp^T Q_{\tau+1}\|_2^2 \ge 1/2$, and $1 - \|\tilde{U}_\perp^T Q_{\tau+2}\|_2^2 \ge 1 - 2\zeta$. Therefore, when $\frac{\log|V|}{2} \ge \frac{\log(\zeta/(1-\|\tilde{U}_\perp^T Q_0\|_2^2))}{\log(1/2\zeta)}$, $\|\tilde{U}_\perp^T Q_{\tau^\star}\|_2 = O\left(\frac{\|A-M\|}{\lambda_K(M)}\right)$. Since $\zeta = o(1)$, to complete this proof, it is sufficient to show that $1 - \|\tilde{U}_\perp^T Q_0\|_2^2 \ge 1/\text{Poly}(n)$ with probability $1 - O(1/n)$, where $\text{Poly}(n)$ is a polynomial function of $n$ with finite order. By Theorem 1.2 of [4] (Please refer to the proof of Lemma 10 of [3]) we can conclude this part with $\text{Poly}(n) = 1/n^4$.

**Step 2.** $\|U_\perp^T \tilde{U}\| = O\left(\frac{\|A-M\|}{\lambda_K(M)}\right)$: We can get an upper bound and a lower bound for $\|AU_\perp\|$ as follows:

$$\|AU_\perp\| = \|(M + A - M)U_\perp\| = \|(A - M)U_\perp\| \le \|A - M\|$$
$$\|AU_\perp\| = \|(\tilde{A} + A - \tilde{A})U_\perp\| \ge \|\tilde{A}U_\perp\| - \|(A - \tilde{A})U_\perp\| \ge \|\tilde{U}\tilde{\Lambda}\tilde{U}^T U_\perp\| - \|A - \tilde{A}\|$$
$$\ge \lambda_K(A)\|\tilde{U}^T U_\perp\| - \|A - \tilde{A}\| \ge (\lambda_K(M) - \|A - M\|)\|\tilde{U}^T U_\perp\| - \|A - M\|.$$

When we combine above bounds, $\|\tilde{U}^T U_\perp\| \le \frac{2\|A-M\|}{(\lambda_K(M)-\|A-M\|)} = O\left(\frac{\|A-M\|}{\lambda_K(M)}\right)$.

### B.8 Proof of Lemma 11

From the definitions of $W$ and $W_\perp$, $Q = WW^T Q + W_\perp W_\perp^T Q$. Since rows of $W$ corresponding to the nodes from the same cluster are the same, the rows of $WW^T Q$ are also the same for the node from the same clusters. Let $WW^T Q(k)$ be the rows of $WW^T Q$ corresponding to $v \in V_k$. Let $\boldsymbol{v}_{(k\ell)} \in \mathbb{R}^{K \times 1}$ such that $k$-th row and $\ell$-th row are $1/\sqrt{|V_k|}$ and $-1/\sqrt{|V_\ell|}$, respectively and other elements are zero. Then, $\|WW^T Q(k) - WW^T Q(\ell)\|^2 = \|W^T Q \boldsymbol{v}_{(k\ell)}\|^2$. Since $\|W^T Q \boldsymbol{x}\| \ge \sqrt{1 - \|W_\perp^T Q\|^2}\|\boldsymbol{x}\|$,

$$\|WW^T Q(k) - WW^T Q(\ell)\|^2 = \Omega\left(\frac{1 - \|W_\perp^T Q\|^2}{n}\right) = \Omega\left(\frac{1}{n}\right) \quad \text{for all } k \ne \ell.$$

Therefore, with some positive $C > 0$,

$$C\frac{|\bigcup_{k,\ell:k\ne\ell} S_k \bigcap V_\ell|}{n} \le \sum_{k,\ell:k\ne\ell} \sum_{v \in S_k \bigcap V_\ell} \|WW^T Q(k) - WW^T Q(\ell)\|^2$$
$$\le 2 \sum_{k,\ell:k\ne\ell} \sum_{v \in S_k \bigcap V_\ell} \|WW^T Q(k) - \xi_{i^\star,k}\|^2 + \|\xi_{i^\star,k} - WW^T Q(\ell)\|^2$$
$$\le 4 \sum_{k,\ell:k\ne\ell} \sum_{v \in S_k \bigcap V_\ell} \|WW^T Q(\ell) - \xi_{i^\star,k}\|^2$$
$$\le 8 \sum_{k,\ell:k\ne\ell} \sum_{v \in S_k \bigcap V_\ell} \|WW^T Q(\ell) - Q_v\|^2 + \|Q_v - \xi_{i^\star,k}\|^2$$
$$\le 8\|W_\perp W_\perp^T Q\|_F^2 + 8r_{i^\star} \le 8K\|W_\perp W_\perp^T Q\|^2 + 8r_{i^\star} \le 8K\|W_\perp^T Q\|^2 + 8r_{i^\star}$$

To conclude this proof, we need to show that $r_{i^\star} = O(\|W_\perp^T Q\|^2)$. Let $i^t$ be an integer between 1 and $\log n$ such that $\frac{\|W_\perp W_\perp^T Q\|_F^2}{n\delta^2} \leq \frac{i^t}{n \log n} \leq \frac{\delta^2}{n}$ with positive constant $\delta$ close to 0. There exists such $i^t$ for any $\delta$, since $\|W_\perp W_\perp^T Q\|^2 = o(1)$ and the rank of $W_\perp W_\perp^T Q$ is $K$. Then,

$$\left| \bigcup_{1 \leq k \leq K} \{v \in V_k : \|Q_v - WW^T Q(k)\|^2 \leq \frac{i^t}{4n \log n}\} \right| \geq n - \|W_\perp W_\perp^T Q\|_F^2 \frac{4n \log n}{i^t}$$

$$\geq n(1 - 4\delta^2).$$

From this, since $\|Q_v - Q_w\|^2 \leq 2\|Q_v - WW^T Q(k)\|^2 + 2\|Q_w - WW^T Q(k)\|^2$, when $v$ satisfying that $\|Q_v - WW^T Q(k)\|^2 \leq \frac{i^t}{4n \log n}$,

$$|X_{i^t,v}| \geq |V_k| - 4\delta^2 n.$$

On the other hand, since $\|Q_v - Q_w\|^2 \geq \frac{1}{2}\|Q_v - WW^T Q(k)\|^2 - \|Q_w - WW^T Q(k)\|^2$, when $v$ satisfying that $\|Q_v - WW^T Q(k)\|^2 \geq \frac{i^t}{n \log n}$,

$$|X_{i,v}| \leq 4\delta^2 n.$$

With small enough constant $\delta$, therefore, when $v$ and $w$ satisfy that $\|Q_v - WW^T Q(k)\|^2 \leq \frac{i^t}{4n \log n}$ and $\|Q_w - WW^T Q(k)\|^2 \geq \frac{i^t}{n \log n}$, $|X_{i,v}| > |X_{i,w}|$, which indicates that the origin of $T_{i^t,k}$ is at least $\|Q_{v_k} - WW^T Q(k)\|^2 \leq \frac{i^t}{n \log n}$ and $|T_{i^t,,k}| \geq |V_k| - 4\delta^2 n$. Since $\|\cdot\|$ is a convex function, by Jensen's inequality, for all $k$,

$$\|WW^T Q(k) - \xi_{i^t,k}\|^2 \leq \frac{\sum_{v \in T_{i^t,k}} \|WW^T Q(k) - Q_v\|^2}{|T_{i^t,k}|} \leq \frac{\|W_\perp W_\perp^T Q\|_F^2}{|V_k| - 4\delta^2 n} = O(\frac{\|W_\perp^T Q\|^2}{n}).$$

Therefore,

$$r_{i^t} = \sum_{k=1}^K \sum_{v \in T_{i^t,k}} \|Q_v - \xi_{i^t,k}\|^2 \leq \sum_{k=1}^K \sum_{v \in V_k} \|Q_v - \xi_{i^t,k}\|^2$$

$$\leq 2 \sum_{k=1}^K \sum_{v \in V_k} \|Q_v - WW^T Q(k)\|^2 + \|WW^T Q(k) - \xi_{i^t,k}\|^2$$

$$\leq 2\|W_\perp W_\perp^T Q\|_F^2 + 2 \sum_{k=1}^K \sum_{v \in V_k} \|WW^T Q(k) - \xi_{i^t,k}\|^2$$

$$= O(\|W_\perp^T Q\|^2) + 2 \sum_{k=1}^K \sum_{v \in V_k} \|WW^T Q(k) - \xi_{i^t,k}\|^2 = O(\|W_\perp^T Q\|^2).$$

Since $r_{i^\star} \leq r_{i^t}$, $r_{i^\star} = O(\|W_\perp^T Q\|^2)$.

## B.9 Proof of Theorem 5

Let $\mu(v, S_k^{(g)}) = \mathbb{E}[\sum_{w \in S_k^{(g)}} A_{vw}]$ and $Var(v, S_k^{(g)}) = \mathbb{E}[(\mu(v, S_k^{(g)}) - \sum_{w \in S_k^{(g)}} A_{vw})^2]$. Since $|\bigcup_{k=1}^K (S_k^{(g)} \setminus V_k^{(g)})| = o(|V^{(g)}|)$ from Theorem 4, $\frac{\mu(v, S_k^{(g)})}{|S_k^{(g)}|} = p(1 + o(1))$ and $\frac{Var(v, S_k^{(g)})}{|S_k^{(g)}|} = p(1 + o(1))$ when $v \in V_k$, and $\frac{\mu(v, S_k^{(g)})}{|S_k^{(g)}|} = q(1 + o(1))$ and $\frac{Var(v, S_k^{(g)})}{|S_k^{(g)}|} = q(1 + o(1))$ when $v \notin V_k$. By Chebyshev's inequality, when $v \in V_k$, $v \in S_k$ with high probability since $\frac{\mu(v, S_k^{(g)}) - \mu(v, S_{k'}^{(g)})}{\sqrt{Var(v, S_k^{(g)})}} = \omega(1)$ for all $k' \neq k$ when $\gamma f(n) = \omega(1)$.

## B.10 Proof of Theorem 6

In this proof, we use Chernoff bound as the form of Lemma 8.1 in [1].

From Theorem 4, Algorithm 1 classifies the arrival nodes at each time block with diminishing fraction of misclassified nodes. Between $S_i^{(\tau)}$ and $S_j^{(\tau+1)}$, the number of connections is $\Theta(B^2 \frac{\bar{f}(n)}{n}) = \Theta(\frac{h^2(n)n}{\min\{f(n), n^{1/3}\} \log^2 n}) = \omega(1)$ from the condition of this theorem. Let $\mu(k, i) = \frac{\sum_{v \in \hat{V}_i} \sum_{w \in S_k^{(\tau)}} A_{wv}}{\sum_{v \in \hat{V}_i} \sum_{w \in S_k^{(\tau)}} 1}$. By the Chernoff bound, with high probability (since $\sum_{v \in \hat{V}_i} \sum_{w \in S_k^{(\tau)}} A_{wv} = \omega(1)$), $\mu(k, i) = p(1 - o(1))$ when $\frac{|S_k^{(\tau)} \cap V_i|}{|S_k^{(\tau)}|} = 1 - o(1)$ and $\mu(k, i) = q(1 + o(1))$ when $\frac{|S_k^{(\tau)} \cap V_i|}{|S_k^{(\tau)}|} = o(1)$. Therefore, with high probability, $S_k^{(\tau)}$ is merged with $\hat{V}_{s(k)}$ such that $\frac{|S_k^{(\tau)} \cap V_{s(k)}|}{|S_k^{(\tau)}|} = 1 - o(1)$. Thus, $\frac{|\hat{V}_k \cap V_k|}{|\hat{V}_k|} = 1 - o(1)$ for all $k$ with high probability.

Since $\frac{|\hat{V}_k \cap V_k|}{|\hat{V}_k|} = 1 - o(1)$ for all $k$, one can easily show using the Chernoff bound that $\frac{N_{v,k}}{|\hat{V}_k|} \geq p(1 - \frac{p-q}{4})$ when $v \in V_k$ and $\frac{N_{v,k'}}{|\hat{V}_{k'}|} \leq q(1 + \frac{p-q}{4})$ when $v \notin V_{k'}$ with probability $1 - O(\exp(-cT \frac{f(n)}{n}))$ with a constant $c > 0$. Thus, the probability for that $\frac{N_{v,k}}{|\hat{V}_k|} \leq \frac{N_{v,k'}}{|\hat{V}_{k'}|}$ for $v \in V_k$ and $k \neq k'$ is $O(\exp(-cT \frac{f(n)}{n}))$.