[Reviews · NeurIPS 2014]

Submitted by Assigned_Reviewer_8

This paper proposes an efficient algorithm for clustering networks. The clustering utilizes the stochastic blockmodel and the authors show that the model can be fitted efficiently with partial data and memory constraints.

Specific Comments:

The version of the stochastic blockmodel reported is a very simple version of the model. Most applications of the stochastic blockmodel use a different intra cluster probability for each cluster and a different inter cluster probability for each pair. Can the blockmodel used herein be used to approximate the more general type of blockmodel?

The results appear to be restricted to the case where p and q have similar structure in terms of n because p=a f(n) and q=b f(n). Could the results be extended to other cases? I suspect that they may hold if p tends to dominate q as n gets large?

The paper would benefit in showing either one or both of the following:
(a) a large but not huge example that shows how a standard stochastic blockmodel fit compares to the fast algorithm presented herein.
(b) the results applied to a huge example (however briefly).
Summary: This paper proposes an efficient algorithm for clustering networks. The clustering utilizes the stochastic blockmodel and the authors show that the model can be fitted efficiently with partial data and memory constraints.

Submitted by Assigned_Reviewer_18

This paper studies community detection of stochastic block models in a streaming setting. Motivated by the need to reduce input data size when the number of nodes is large, the author(s) first considered a subsampling technique, where community detection is carried out on a random sample of columns is obtained from the adjacency matrix. The main result is a sharp threshold on the sampling rate. It is shown that below this critical rate, no method can recover the communities with vanishing error rate; and a modified spectral method is provided and proved to be consistent when the sampling rate exceeds the threshold. This result is further used to construct a sub-linear memory streaming algorithm that is consistent.

This paper addresses an important problem, and is clearly written. The theoretical results are sharp and solid. The algorithms are modifications of previous spectral methods, and their practicality is not obvious and is not demonstrated through data examples.
Summary: This paper studies network community detection under the subsample and streaming context. It is an early effort in such a topic and is likely to make a good impact.

Submitted by Assigned_Reviewer_24

The paper constructs and analyzes SBM estimation algorithms for two settings: partial information (where only some columns of the adjacency matrix are observed), and streaming (where columns are observed one at a time). Information-theoretic lower bounds are established, and an asymptotic analysis of the estimation algorithms is conducted. No simulation study was provided, which I find concerning since some of the algorithms have difficult-to-set tuning parameters.

Quality:
I am not an expert on SBM theory, so I cannot comment on the quality of the proofs - but I did find the presentation to be accessible enough for my level of expertise. The theory appears quite complete, with information-theoretic lower bounds nicely matched by the algorithm upper bounds. I also found it nice how Algorithms 3/4 are simply extensions of 1/2, thus making an explicit connection between the streaming setting and partial-information setting.

My major concern is the lack of simulation study, and the fact that Algorithms 3 and 4 require tuning parameters (which I worry they will be highly sensitive to). While I am convinced that Algorithms 1 and 2 will work in practice, I cannot say the same about Algorithms 3 and 4.

Clarity:
Well-written in most places, but there are some paragraphs of noticeably lower quality, that are obviously written by a different person.

The writing did not make the difference between Algorithms 3 (offline streaming) and 4 (online streaming) clear - it looks like the information is there, but scattered throughout Section 4. The authors should provide a summary paragraph stating exactly how those algorithms differ.

Originality:
I only have superficial knowledge of the theoretical SBM literature, but this paper addresses settings (partial information; streaming) that are atypical in my experience. I would say this paper is original.

Significance:
To the best of my knowledge, I am not aware of SBM estimation algorithms for the partial information or streaming settings. I would consider this paper to be highly significant, had the authors shown via simulation that the algorithms were actually practical to implement and tune.

Minor comments:
-On page 8, the authors claim that the supplementary material contains a precise statement on the performance guarantees on their methods vs streaming PCA. I could not find this statement in the supplementary.
-The abstract does not make it clear that the paper's contribution is solely theoretical, with not even a simulation study. If the paper is accepted as-is, its scope should be made clear in the abstract.
Summary: The paper provides valuable insight into the task of network clustering, under the SBM model with either limited information about the network, or in the streaming setting. However, the lack of even a basic simulation study, coupled with the presence of tuning parameters in Algorithms 3 and 4, lead me to question if the presented algorithms are actually practical.

Submitted by Assigned_Reviewer_45

The paper describes two related algorithms for finding clusters under the stochastic block model assumption when there is partial information (in the sense that only a fraction of the columns of the adjacency matrix are revealed) or in a data streaming setting (where fewer than the size of the input bits are retained to perform the computation).

Pros:

Another contribution to an area of recent interest, both with respect to data-constrained computation, clustering matrices, etc.

The main algorithm splits the problem into two steps, which is probably the correct way to do streaming clustering under model assumptions like this stochastic block model. Making that explicit highlights both important points.

Cons:

The algorithm is of theoretical interest, as it makes unrealistically strong assumptions for realistic networks of even moderate size and since it doesn't work at important "boundaries" of edge density, e.g., extremely sparse graphs.

Claims with respect to how this work fits into the streaming and community literature are very much overstated, as mentioned below.

The description of the algorithm seems to not make clear what I think is going on: that under the stochastic block model with large blocks, then input has very nice structure, and so the main goal of the algorithm is to make sure that the sampling doesn't overly concentrate certain quantities in the "empirical" estimates. For example, I think that is what the trimming step is doing, and I think that is similar to what Feige and others, Montaneri and others, and probably others have done.

A few other comments.

(1) The problem as stated is unrealistic for what most people would consider community detection in realistic networks. For example, the clusters don't overlap, and the results don't go through when the graphs are extremely sparse (which is typical) or extremely dense (which may arise occasionally). Thus, they are best viewed as contributing to the recent work on spectral-like algorithms for stochastic block (which also suffer from similar drawbacks) and related models under various computational constraints (which is of interest).

(2) No empirical evaluation. This isn't necessarily a bad thing, since if there were it would likely be trivial (in the sense that they would simply show a phase transition which they establish) or overly-idealized (i.e., they would have to "simulate" memory limitations, since dealing with realistic memory management is very nontrivial).

(3) The columns are revealed one by one. This is a rather strong assumption in terms of motivations, and it means that it is much easier to obtain much stronger results than if elements or blocks are revealed. For example, much stronger results are obtained in low-rank matrix approximation when columns are sampled than when elements are sampled. More formal data streaming models, e.g., in the theoretical computer science literature consider other data presentation formats.

(4) Finally, if the authors don't know of other related work on clustering or partitioning or community finding, it is for lack of effort. For example, a few quick web searches find many papers, including the following papers:

FENNEL: Streaming Graph Partitioning for Massive Scale Graphs, by Tsourakaki, et al.

Online Analysis of Community Evolution in Data Streams, by Aggarwal and Yu

Streaming Graph Partitioning for Large Distributed Graphs, by Stanton and Kliot

Sparse Cut Projections in Graph Streams, by Das Sarma, Gollapudi and Panigrahy

These papers are all sufficiently different than the present paper that there is novelty in the present paper and no overlap. But they illustrate that it much more accurate to describe the present paper as an interesting but relatively minor improvement on recent work on stochastic blockmodeling under various well-motivated memory constraints than as the first community detection algorithms in the data stream model or as a paper that introduces the problem of community detection with partial information, both of which are claimed but neither of which is correct.
Summary: Overall a reasonable paper on a topic of interest. If the paper is accepted, I suggest that the authors adjust their claims to be more modest and correct and that they also work on the presentation to highlight their few nice contributions.
Author Feedback
Author rebuttal: We thank the reviewers for their interesting and constructive reviews that in turn will help us to clarify the contributions and improve the manuscript.

REVIEWER #18.

1) Numerical experiments: In addition to the theoretical results already reported in the paper, we made preliminary simulation results illustrating the benefits achieved by the proposed streaming algorithms: for large networks (> million of nodes), our algorithms detect communities while this is not possible using non-streaming spectral algorithms (we run the experiments on a standard PC whose configuration will be described). This initial numerical result will be included in the paper.

REVIEWER #24.

2) The readability of the paper can be significantly improved, and we will make an important effort to polish the manuscript. We will in particular emphasize the differences between Algorithms 3 and 4.
3) Numerical experiments: see point 1 above.
4) Tuning parameters for Algorithms 3 and 4: as noted by the reviewer, the density of the network p is given as an input and in practice, it needs to be estimated. Each column of the matrix allows us to compute the degree of the corresponding node so that we can estimate p by averaging the observed degrees. To simplify the analysis of Algorithms 3 and 4, we decided to set p as an input parameter. The other parameter of Algorithm 3 is h(n) which determines the size of the blocks. This parameter needs NOT to be tuned according to the incoming data. The larger the blocks are the better the algorithm performs. The only limiting factor to determine h(n) is memory. Theorem 6 indicates how to tune h(n): if we want an accurate cluster detection, we should set h(n) so that it scales as M/n where M is the available memory. The same analysis hold for the parameter m(n) in Algorithm 4. We will clarify these points in the paper.
5) Minor comments: the performance of the proposed method vs. that of streaming PCA algorithm will be clarified in the supplementary material (currently the performance comparison is presented in Remark 15, which requires clarification). We will rewrite the abstract the highlight the contributions.

REVIEWER #45

6) Graphs with moderate sizes and very sparse networks: Indeed, our results do not consider networks with moderate sizes or very sparse networks. Our objective was to identify necessary and sufficient conditions for asymptotically accurate cluster detection (the proportion of misclassified nodes vanishes as the network size grows large). For networks with moderate sizes and very sparse networks, accurate detection is not achievable for the SBM. For example, for very sparse graphs, the proportion of isolated nodes (these nodes cannot be classified) does not vanish, as the network grows large.
7) Trimming step: The trimming step in our algorithms is classical in spectral methods and the reviewer has the right interpretation about its use. However our algorithms include novel and crucial steps (such as the idea of exploiting indirect edges).
8) Numerical experiments: see point 1 above. We agree with the reviewer: evaluating precisely the memory used by the algorithms in practice is rather difficult and is left for future work.
9) Columns revealed one-by-one: The idea of considering such a model originates from the way online social services work: users generate requests to the service in a sequential manner, and each time a request is made, the information about the corresponding user is revealed. We are thinking to extend the results to scenarios where individual elements or blocks of the matrix are sequentially revealed.
10) Related work: We restricted the list of references to papers dealing with SBM-like models, i.e., statistical models. We know very well the literature related to the 'FENNEL' algorithm and 'Streaming Graph Partitioning...' by Stanton and Kliot. Their main motivation is to store a huge graph on several servers in order to make efficient parallel computing. The measure of performance (load balancing and minimizing messages among servers) is quite different from ours and their analysis is mainly based on simulations. The paper of Aggrawal and Yu discusses the problem of online change detection which we do not address. Finally the paper 'Sparse Cut Projections in Graph Streams' is a theoretical CS approach to the problem where worst case performances are given in term of conductance and sparsity of the graph. Such bounds are useless in our setting. We will carefully state how our paper fits into the streaming and clustering literature.

REVIEWER #8

11) Models: We use a simple model for the generation of clusters (fixed intra- and inter-cluster connection probabilities). The results are however not limited to this model and most of the results can be extended to more general scenarios (e.g. with different intra- and inter-cluster connection probabilities).
12) As the reviewer pointed out, the results still hold when p dominates q (q/p tends to 0 as n goes to infinity). We chose to scale p and q with the same function f(n) to get simple and clear conditions in our main theorems.
13) Numerical experiments: see point 1 above.